# LEARNING ENERGY-BASED MODELS BY COOPERATIVE DIFFUSION RECOVERY LIKELIHOOD

**Yaxuan Zhu**
UCLA
yaxuanzhu@g.ucla.edu

**Jianwen Xie**
Akool Research
jianwen@ucla.edu

**Ying Nian Wu**
UCLA
ywu@stat.ucla.edu

**Ruiqi Gao**
Google DeepMind
ruiqig@google.com

## ABSTRACT

Training energy-based models (EBMs) on high-dimensional data can be both challenging and time-consuming, and there exists a noticeable gap in sample quality between EBMs and other generative frameworks like GANs and diffusion models. To close this gap, inspired by the recent efforts of learning EBMs by maximizing diffusion recovery likelihood (DRL), we propose cooperative diffusion recovery likelihood (CDRL), an effective approach to tractably learn and sample from a series of EBMs defined on increasingly noisy versions of a dataset, paired with an initializer model for each EBM. At each noise level, the two models are jointly estimated within a cooperative training framework: samples from the initializer serve as starting points that are refined by a few MCMC sampling steps from the EBM. The EBM is then optimized by maximizing recovery likelihood, while the initializer model is optimized by learning from the difference between the refined samples and the initial samples. In addition, we made several practical designs for EBM training to further improve the sample quality. Combining these advances, our approach significantly boost the generation performance compared to existing EBM methods on CIFAR-10 and ImageNet datasets. We also demonstrate the effectiveness of our models for several downstream tasks, including classifier-free guided generation, compositional generation, image inpainting and out-of-distribution detection.

## 1 INTRODUCTION

Energy-based models (EBMs), as a class of probabilistic generative models, have exhibited their flexibility and practicality in a variety of application scenarios, such as realistic image synthesis (Xie et al., 2016; 2018a; Nijkamp et al., 2019; Du & Mordatch, 2019; Arbel et al., 2021; Hill et al., 2022; Xiao et al., 2021; Lee et al., 2023; Grathwohl et al., 2021; Cui & Han, 2023), graph generation (Liu et al., 2021), compositional generation (Du et al., 2020; 2023), video generation (Xie et al., 2021c), 3D generation (Xie et al., 2021a; 2018b), simulation-based inference (Glaser et al., 2022), stochastic optimization (Kong et al., 2022), out-of-distribution detection (Grathwohl et al., 2020; Liu et al., 2020), and latent space modeling (Pang et al., 2020; Zhu et al., 2023; Zhang et al., 2023; Yu et al., 2023). Despite these successes of EBMs, training and sampling from EBMs remains challenging, mainly due of the intractability of the partition function in the distribution.

Recently, Diffusion Recovery Likelihood (DRL) (Gao et al., 2021) has emerged as a powerful framework for estimating EBMs. Inspired by diffusion models (Sohl-Dickstein et al., 2015; Ho et al., 2020; Song & Ermon, 2019), DRL assumes a sequence of EBMs for the marginal distributions of samples perturbed by a Gaussian diffusion process, where each EBM is trained with recovery likelihood that maximizes the conditional probability of the data at the current noise level given their noisy versions at a higher noise level. Compared to the regular likelihood, Maximizing recovery likelihood is more tractable, as sampling from the conditional distribution is much easier than sampling from the marginal distribution. DRL achieves exceptional generation performance among EBM-based generative models. However, a noticeable performance gap still exists between the sample quality of EBMs and other generative frameworks like GANs or diffusion models. Moreover, DRL requires around 30 MCMC sampling steps at each noise level to generate valid samples, which can be time-consuming during both training and sampling processes.

To further close the performance gap and expedite EBM training and sampling with fewer MCMC sampling steps, we introduce Cooperative Diffusion Recovery Likelihood (CDRL), that jointly estimates a sequence of EBMs and MCMC initializers defined on data perturbed by a diffusion process. At each noise level, the initializer and EBM are updated by a cooperative training scheme (Xie et al., 2018a): The initializer model proposes initial samples by predicting the samples at the current noise level given their noisy versions at a higher noise level. The initial samples are then refined by a few MCMC sampling steps from the conditional distribution defined by the EBM. Given the refined samples, the EBM is updated by maximizing recovery likelihood, and the initializer is updated to absorb the difference between the initial samples and the refined samples. The introduced initializer models learn to accumulate the MCMC transitions of the EBMs, and reproduce them by direct ancestral sampling. Combining with a new noise schedule and a variance reduction technique, we achieve significantly better performance than the existing methods of estimating EBMs. We further incorporate classifier-free guidance (CFG) (Ho & Salimans, 2022) to enhance the performance of conditional generation, and we observe similar trade-offs between sample quality and sample diversity as CFG for diffusion models when adjusting the guidance strength. In addition, we showcase that our approach can be applied to perform several useful downstream tasks, including compositional generation, image inpainting and out-of-distribution detection.

Our main contributions are as follows: (1) We propose cooperative diffusion recovery likelihood (CDRL) that tractably and efficiently learns and samples from a sequence of EBMs and MCMC initializers; (2) We make several practical design choices related to noise scheduling, MCMC sampling, noise variance reduction for EBM training; (3) Empirically we demonstrate that CDRL achieves significant improvements on sample quality compared to existing EBM approaches, on CIFAR-10 and ImageNet $32 \times 32$ datasets; (4) We show that CDRL has great potential to enable more efficient sampling with sampling adjustment techniques; (5) We demonstrate CDRL's ability in compositional generation, image inpainting and out-of-distribution (OOD) detection, as well as its compatibility with classifier-free guidance for conditional generation.

## 2    PRELIMINARIES ON ENERGY-BASED MODELS

Let $\mathbf{x} \sim p_{\mathrm{data}}(\mathbf{x})$ be a training example from an underlying data distribution. An energy-based model defines the density of $\mathbf{x}$ by

$$p_\theta(\mathbf{x}) = \frac{1}{Z_\theta} \exp(f_\theta(\mathbf{x})), \tag{1}$$

where $f_\theta$ is the unnormalized log density, or negative energy, parametrized by a neural network with a scalar output. $Z_\theta$ is the normalizing constant or partition function. The derivative of the log-likelihood function of an EBM can be approximately written as

$$\mathcal{L}'(\theta) = \mathbb{E}_{p_{\mathrm{data}}} \left[ \frac{\partial}{\partial \theta} f_\theta(\mathbf{x}) \right] - \mathbb{E}_{p_\theta} \left[ \frac{\partial}{\partial \theta} f_\theta(\mathbf{x}) \right], \tag{2}$$

where the second term is analytically intractable and has to be estimated by Monte Carlo samples from the current model $p_\theta$. Therefore, applying gradient-based optimization for an EBM usually involves an inner loop of MCMC sampling, which can be time-consuming for high-dimensional data.

## 3    COOPERATIVE DIFFUSION RECOVERY LIKELIHOOD

### 3.1    DIFFUSION RECOVERY LIKELIHOOD

Given the difficulty of sampling from the marginal distribution $p(\mathbf{x})$ defined by an EBM, we could instead estimate a sequence of EBMs defined on increasingly noisy versions of the data and jointly estimate them by maximizing *recovery likelihood*. Specifically, assume a sequence of noisy training examples perturbed by a Gaussian diffusion process: $\mathbf{x}_0, \mathbf{x}_1, ..., \mathbf{x}_T$ such that $\mathbf{x}_0 \sim p_{\mathrm{data}}$; $\mathbf{x}_{t+1} = \alpha_{t+1}\mathbf{x}_t + \sigma_{t+1}\epsilon$. Denote $\mathbf{y}_t = \alpha_{t+1}\mathbf{x}_t$ for notation simplicity. The marginal distributions of $\{\mathbf{y}_t; t = 1, ..., T\}$ are modeled by a sequence of EBMs: $p_\theta(\mathbf{y}_t) = \frac{1}{Z_{\theta,t}} \exp(f_\theta(\mathbf{y}_t; t))$. Then the conditional EBM of $\mathbf{y}_t$ given the sample $\mathbf{x}_{t+1}$ at a higher noise level can be derived as

$$p_\theta(\mathbf{y}_t | \mathbf{x}_{t+1}) = \frac{1}{\tilde{Z}_{\theta,t}(\mathbf{x}_{t+1})} \exp \left( f_\theta(\mathbf{y}_t; t) - \frac{1}{2\sigma_{t+1}^2} \|\mathbf{y}_t - \mathbf{x}_{t+1}\|^2 \right), \tag{3}$$

where $\tilde{Z}_{\theta,t}(\mathbf{x}_{t+1})$ is the partition function of the conditional EBM dependent on $\mathbf{x}_{t+1}$. Compared with the marginal EBM $p_\theta(\mathbf{y}_t)$, when $\sigma_{t+1}$ is small, the extra quadratic term in $p_\theta(\mathbf{y}_t|\mathbf{x}_{t+1})$ constrains the conditional energy landscape to be localized around $\mathbf{x}_{t+1}$, making the latter less multi-modal and easier to sample from with MCMC. In the extreme case when $\sigma_{t+1}$ is infinitesimal, $p_\theta(\mathbf{y}_t|\mathbf{x}_{t+1})$ is approximately a Gaussian distribution that can be tractably sampled from and has a close connection to diffusion models (Gao et al., 2021). In the other extreme case when $\sigma_{t+1} \to \infty$, the conditional distribution falls back to the marginal distribution, and we lose the advantage of being more MCMC friendly for the conditional distribution. Therefore, we need to maintain a small $\sigma_{t+1}$ between adjacent time steps, and to equip the model with the ability of generating new samples from white noises, we end up with estimating a sequence of EBMs defined on the diffusion process. We use the variance-preserving noise schedule (Song et al., 2021b), under which case we have $\mathbf{x}_t = \bar{\alpha}_t\mathbf{x}_0 + \bar{\sigma}_t\boldsymbol{\epsilon}$, where $\bar{\alpha}_t = \prod_{t=1}^{T} \alpha_t$ and $\bar{\sigma}_t = \sqrt{1 - \bar{\alpha}_t^2}$.

We estimate each EBM by maximizing the following recovery log-likelihood function at each noise level (Bengio et al., 2013):

$$\mathcal{J}_t(\theta) = \frac{1}{n}\sum_{i=1}^{n} \log p_\theta(\mathbf{y}_{t,i}|\mathbf{x}_{t+1,i}), \tag{4}$$

where $\{\mathbf{y}_{t,i}, \mathbf{x}_{t+1,i}\}$ are pair of samples at time steps $t$ and $t+1$. Sampling from $p_\theta(\mathbf{y}_t|\mathbf{x}_{t+1})$ can be achieved by running $K$ steps of Langevin dynamics from the initialization point $\tilde{\mathbf{y}}_t^0 = \mathbf{x}_{t+1,i}$ and iterating

$$\tilde{\mathbf{y}}_t^{\tau+1} = \tilde{\mathbf{y}}_t^\tau + \frac{s_t^2}{2}\left(\nabla_{\mathbf{y}} f_\theta(\tilde{\mathbf{y}}_t^\tau; t) - \frac{1}{\sigma_{t+1}^2}(\tilde{\mathbf{y}}_t^\tau - \mathbf{x}_{t+1})\right) + s_t\boldsymbol{\epsilon}^\tau, \tag{5}$$

where $s_t$ is the step size and $\tau$ is the index of MCMC sampling step. With the samples, the updating of EBMs then follows the same learning gradients as MLE (Equation 2), as the extra quadratic term $-\frac{1}{2\sigma_{t+1}^2}\|\mathbf{y}_t - \mathbf{x}_{t+1}\|^2$ in $p_\theta(\mathbf{y}_t|\mathbf{x}_{t+1})$ does not involve learnable parameters. It is worth noting that maximizing recovery likelihood still guarantees an unbiased estimator of the true parameters of the *marginal distribution* of the data.

## 3.2 Amortizing MCMC sampling with initializer models

Although $p_\theta(\mathbf{y}_t|\mathbf{x}_{t+1})$ is easier to sample from than $p_\theta(\mathbf{y}_t)$, when $\sigma_{t+1}$ is not infinitesimal, the initialization of MCMC sampling, $\mathbf{x}_{t+1}$, may still be far from the data manifold of $\mathbf{y}_t$. This necessitates a certain amount of MCMC sampling steps at each noise level (e.g., 30 steps of Langevin dynamics in Gao et al. (2021)). Naively reducing the number of sampling steps would lead to training divergence or performance degradation.

To address this issue, we propose to learn an initializer model jointly with the EBM at each noise level, which maps $\mathbf{x}_{t+1}$ closer to the manifold of $\mathbf{y}_t$. Our work is inspired by the CoopNets (Xie et al., 2018a; 2021b; 2022b), which shows that jointly training a top-down generator via MCMC teaching will help the training of a single EBM model. We take this idea and generalize it to the recovery-likelihood model. More discussions are included in Appendix A. Specifically, the initializer model at noise level $t$ is defined as

$$q_\phi(\mathbf{y}_t|\mathbf{x}_{t+1}) \sim \mathcal{N}(\boldsymbol{g}_\phi(\mathbf{x}_{t+1}; t), \tilde{\sigma}_t^2\boldsymbol{I}). \tag{6}$$

It serves as a coarse approximation to $p_\theta(\mathbf{y}_t|\mathbf{x}_{t+1})$, as the former is a single-mode Gaussian distribution while the latter can be multi-modal. A more general formulation would be to involve latent variables $\mathbf{z}_t$ following a certain simple prior $p(\mathbf{z}_t)$ into $\boldsymbol{g}_\phi$. Then $q_\phi(\mathbf{y}_t, t|\mathbf{x}_{t+1}) = \mathbb{E}_{p(\mathbf{z}_t)}[q_\phi(\mathbf{y}_t, \mathbf{z}_t, t|\mathbf{x}_{t+1})]$ can be non-Gaussian (Xiao et al., 2022). However, we empirically find that the simple initializer in Equation 6 works well. Compared with the more general formulation, the simple initializer avoids the inference of $\mathbf{z}_t$ which may again require MCMC sampling, and leads to more stable training. Different from Xiao et al. (2022), samples from the initializer just serves as the starting points and are refined by sampling from the EBM, instead of being treated as the final samples. We follow (Ho et al., 2020) to set $\tilde{\sigma}_t = \sqrt{\frac{1-\bar{\alpha}_t^2}{1-\bar{\alpha}_{t+1}^2}}\sigma_t$. If we treat the sequence of initializers as the reverse process, such choice of $\tilde{\sigma}_t$ corresponds to the lower bound of the standard deviation given by $p_{\text{data}}$ being a delta function (Sohl-Dickstein et al., 2015).

### 3.3 COOPERATIVE TRAINING

We jointly train the sequence of EBMs and intializers in a cooperative fashion. Specifically, at each iteration, for a randomly sampled noise level $t$, we obtain an initial sample $\hat{\mathbf{y}}_t$ from the intializer model. Then a synthesized sample $\tilde{\mathbf{y}}_t$ from $p(\mathbf{y}_t|\mathbf{x}_{t+1})$ is generated by initializing from $\hat{\mathbf{y}}_t$ and running a few steps of Langevin dynamics (Equation 5). The parameters of EBM are then updated by maximizing the recovery log-likelihood function (Equation 4). The learning gradient of EBM is

$$\nabla_\theta \mathcal{J}_t(\theta) = \nabla_\theta \left[ \frac{1}{n} \sum_{i=1}^n f_\theta(\mathbf{y}_{t,i}; t) - \frac{1}{n} \sum_{i=1}^n f_\theta(\tilde{\mathbf{y}}_{t,i}; t) \right]. \tag{7}$$

To train the intializer model that amortizes the MCMC sampling process, we treat the revised sample $\tilde{\mathbf{y}}_t$ by the EBM as the observed data of the initializer model, and estimate the parameters of the initializer by maximizing log-likelihood:

$$\mathcal{L}_t(\phi) = \frac{1}{n} \sum_{i=1}^n \left[ -\frac{1}{2\tilde{\sigma}_t^2} \|\tilde{\mathbf{y}}_{t,i} - \boldsymbol{g}_\phi(\mathbf{x}_{t+1,i}; t)\|^2 \right]. \tag{8}$$

That is, the initializer model learns to absorb the difference between $\hat{\mathbf{y}}_t$ and $\tilde{\mathbf{y}}_t$ at each iteration so that $\hat{\mathbf{y}}_t$ is getting closer to the samples from $p_\theta(\mathbf{y}_t|\mathbf{x}_{t+1})$. In practice, we re-weight $\mathcal{L}_t(\phi)$ across different noise levels by removing the coefficient $\frac{1}{2\tilde{\sigma}_t^2}$, similar to the "simple loss" in diffusion models. The training algorithm is summarized in Algorithm 1.

After training, we generate new samples by starting from Gaussian white noise and progressively samples $p_\theta(\mathbf{y}_t|\mathbf{x}_{t+1})$ at decreasingly lower noise levels. For each noise level, an initial proposal is generated from the intializer model, followed by a few steps of Langevin dynamics from the EBM. See Algorithm 2 for a summary.

### 3.4 NOISE VARIANCE REDUCTION

We further propose a simple way to reduce the variance of training gradients. In principle, the pair of $\mathbf{x}_t$ (or $\mathbf{y}_t$) and $\mathbf{x}_{t+1}$ is generated by $\mathbf{x}_t \sim \mathcal{N}(\bar{\alpha}_t \mathbf{x}_0, \bar{\sigma}_t^2 \boldsymbol{I})$ and $\mathbf{x}_{t+1} \sim \mathcal{N}(\alpha_{t+1} \mathbf{x}_t, \sigma_{t+1}^2 \boldsymbol{I})$. Alternatively, we can fix the Gaussian white noise $\mathbf{e} \sim \mathcal{N}(0, \boldsymbol{I})$, and sample pair $(\mathbf{x}_t', \mathbf{x}_{t+1}')$ by

$$\mathbf{x}_t' = \bar{\alpha}_t \mathbf{x}_0 + \bar{\sigma}_t \mathbf{e}$$
$$\mathbf{x}_{t+1}' = \bar{\alpha}_{t+1} \mathbf{x}_t' + \bar{\sigma}_{t+1} \mathbf{e}. \tag{9}$$

In other words, both $\mathbf{x}_t'$ and $\mathbf{x}_{t+1}'$ are linear interpolation between the clean sample $\mathbf{x}_0$ and a sampled white noise image $\mathbf{e}$. $\mathbf{x}_t'$ and $\mathbf{x}_{t+1}'$ have the same marginal distributions as $\mathbf{x}_t$ and $\mathbf{x}_{t+1}$. But $\mathbf{x}_t'$ is deterministic given $\mathbf{x}_0$ and $\mathbf{x}_{t+1}'$, while there's still variance for $\mathbf{x}_t$ given $\mathbf{x}_0$ and $\mathbf{x}_{t+1}$. This schedule is related to the ODE forward process used in flow matching (Lipman et al., 2022) and rectified flow (Liu et al., 2022b).

### 3.5 CONDITIONAL GENERATION AND CLASSIFIER-FREE GUIDANCE

Ho & Salimans (2022) introduced classifier-free guidance, which greatly enhances the sample quality of conditional diffusion models and balances between sample quality and diversity by adjusting the strength of guidance. Given the close connection between EBMs and diffusion models, we show that it is possible to apply classifier-free guidance in our CDRL as well. Specifically, suppose $c$ is the context (e.g., a label or a text description). At each noise level we jointly estimate an unconditional EBM $p_\theta(\mathbf{y}_t) \propto \exp(f_\theta(\mathbf{y}_t; t))$ and a conditional EBM $p_\theta(\mathbf{y}_t|c) \propto \exp(f_\theta(\mathbf{y}_t; c, t))$. By following the classifier-free guidance (Ho & Salimans, 2022), we can perform conditional sampling using the following linear combination of the conditional and unconditional gradients:

$$\nabla_\mathbf{y} \log \tilde{p}_\theta(\mathbf{y}_t|c) = (w+1)\nabla_\mathbf{y} \log p_\theta(\mathbf{y}_t; c, t) - w\nabla_\mathbf{y} \log p_\theta(\mathbf{y}_t; t) \tag{10}$$
$$= (w+1)\nabla_\mathbf{y} f_\theta(\mathbf{y}_t; c, t) - w\nabla_\mathbf{y} f_\theta(\mathbf{y}_t; t), \tag{11}$$

where $w$ controls the guidance strength. We adopt a single neural network to parameterize both conditional and unconditional models, where for the unconditional model we can simply input a null token

---

**Algorithm 1** CDRL Training

---

**Input**: (1) observed data $\mathbf{x}_0 \sim p_{\text{data}}(\mathbf{x})$; (2) Number of noise levels $T$; (3) Number of Langevin sampling steps $K$ per noise level; (4) Langevin step size at each noise level $s_t$; (5) Learning rate $\eta_\theta$ for EBM $f_\theta$; (6) Learning rate $\eta_\phi$ for initializer $g_\phi$;
**Output**: Parameters $\theta, \phi$
   Randomly initialize $\theta$ and $\phi$.
   **repeat**
      Sample noise level $t$ from $\{0, 1, ..., T - 1\}$.
      Sample $\boldsymbol{\epsilon} \sim \mathcal{N}(0, \boldsymbol{I})$. Let $\mathbf{x}_{t+1} = \bar{\alpha}_{t+1}\mathbf{x}_0 + \bar{\sigma}_{t+1}\boldsymbol{\epsilon}$, $\mathbf{y}_t = \alpha_{t+1}(\bar{\alpha}_t\mathbf{x}_0 + \bar{\sigma}_t\boldsymbol{\epsilon})$.
      Generate the initial sample $\hat{\mathbf{y}}_t$ following Equation 6.
      Generate the refined sample $\mathbf{y}_t$ by running $K$ steps of Langevin dynamics starting from $\hat{\mathbf{y}}_t$ following Equation 5.
      Update EBM parameter $\theta$ following the gradients in Equation 7.
      Update initializer parameter $\phi$ by maximizing Equation 8.
   **until** converged

---

**Algorithm 2** CDRL Sampling

---

**Input**: (1) Number of noise levels $T$; (2) Number of Langevin sampling steps $K$ at each noise level; (3) Langevin step size at each noise level $\delta_t$; (4) Trained EBM $f_\theta$; (5) Trained initializer $g_\phi$;
**Output**: Samples $\tilde{\mathbf{x}}_0$
   Randomly initialize $\mathbf{x}_T \sim \mathcal{N}(0, \boldsymbol{I})$.
   **for** $t = T - 1$ **to** $0$ **do**
      Generate initial proposal $\hat{\mathbf{y}}_t$ following Equation 6.
      Update $\hat{\mathbf{y}}_t$ to $\tilde{\mathbf{y}}_t$ by $K$ iterations of Equation 5.
      Let $\tilde{\mathbf{x}}_t = \tilde{\mathbf{y}}_t/\alpha_{t+1}$.
   **end for**

---

$\varnothing$ for the class condition $c$ when outputing the negative energy, i.e. $f_\theta(\mathbf{y}_t; t) = f_\theta(\mathbf{y}_t; c = \varnothing, t)$. Similarly, for the initializer model, we jointly estimate an unconditional model $q_\phi(\mathbf{y}_t|\mathbf{x}_{t+1}) \sim \mathcal{N}(\boldsymbol{g}_\phi(\mathbf{x}_{t+1}; t), \tilde{\sigma}_t^2\boldsymbol{I})$ and a conditional model $q_\phi(\mathbf{y}_t|c, \mathbf{x}_{t+1}) \sim \mathcal{N}(\boldsymbol{g}_\phi(\mathbf{x}_{t+1}; c, t), \tilde{\sigma}_t^2\boldsymbol{I})$. We parameterize both models in a single neural network. Since both models follow Gaussian distributions, the scaled conditional distribution with classifier-free guidance is still a Gaussian distribution (Dhariwal & Nichol, 2021):

$$\tilde{q}_\phi(\mathbf{y}_t|c, \mathbf{x}_{t+1}) = \mathcal{N}\left((w + 1)\boldsymbol{g}_\phi(\mathbf{x}_{t+1}; c, t) - w\boldsymbol{g}_\phi(\mathbf{x}_{t+1}; t), \tilde{\sigma}_t^2\boldsymbol{I}\right). \tag{12}$$

### 3.6 COMPOSITIONALITY IN ENERGY-BASED MODEL

One attractive property of EBMs is compositionality: one can combine multiple EBMs conditioned on individual concepts, and re-normalize it to create a new distribution conditioned on the intersection of those concepts. Specifically, considering two EBMs $p_\theta(\mathbf{x}|c_1) \propto \exp(f_\theta(\mathbf{x}; c_1))$ and $p_\theta(\mathbf{x}|c_2) \propto \exp(f_\theta(\mathbf{x}; c_2))$ that are conditioned on two separate concepts $c_1$ and $c_2$ respectively, Du et al. (2020); Lee et al. (2023) construct a new EBM conditioned on both concepts as $p_\theta(\mathbf{x}|c_1, c_2) \propto \exp(f_\theta(\mathbf{x}; c_1) + f_\theta(\mathbf{x}; c_2))$ based on the production of expert (Hinton, 2002). Specifically, suppose the two concepts $c_1$ and $c_2$ are conditionally independent given the observed example $\mathbf{x}$. Then we have

$$\begin{aligned}
\log p_\theta(\mathbf{x}|c_1, c_2) &= \log p_\theta(c_1, c_2|\mathbf{x}) + \log p_\theta(\mathbf{x}) + \text{const.} \\
&= \log p_\theta(c_1|\mathbf{x}) + \log p_\theta(c_2|\mathbf{x}) + \log p_\theta(\mathbf{x}) + \text{const.} \\
&= \log p_\theta(\mathbf{x}|c_1) + \log p_\theta(\mathbf{x}|c_2) - \log p_\theta(\mathbf{x}) + \text{const.},
\end{aligned}$$

where const. is a constant term independent of $\mathbf{x}$. The composition can be generalized to include an arbitrary number of concepts. Suppose we have $M$ conditionally independent concepts, then

$$\log p_\theta(\mathbf{x}|c_i, i = 1, ..., M) = \sum_{i=1}^{M} \log p_\theta(\mathbf{x}|c_i) - (M - 1)\log p_\theta(\mathbf{x}) + \text{const.} \tag{13}$$

We can combine the compositional log-density (Equation 13) with classifier-free guidance (Equation 11) to further improve the alignment of generated samples with given concepts. The sampling

gradient of the scaled log-density function is given by

$$\nabla_{\mathbf{x}} \log \tilde{p}_\theta(\mathbf{x}|c_i, i=1,...,M) = \nabla_{\mathbf{x}} \left[ (w+1)\sum_{i=1}^{M} \log p_\theta(\mathbf{x}|c_i) - (Mw+M-1)\log p_\theta(\mathbf{x}) + \text{const} \right]$$

$$= (w+1)\sum_{i=1}^{M} \nabla_{\mathbf{x}} f_\theta(\mathbf{x}|c_i) - (Mw+M-1)\nabla_{\mathbf{x}} f_\theta(\mathbf{x}). \tag{14}$$

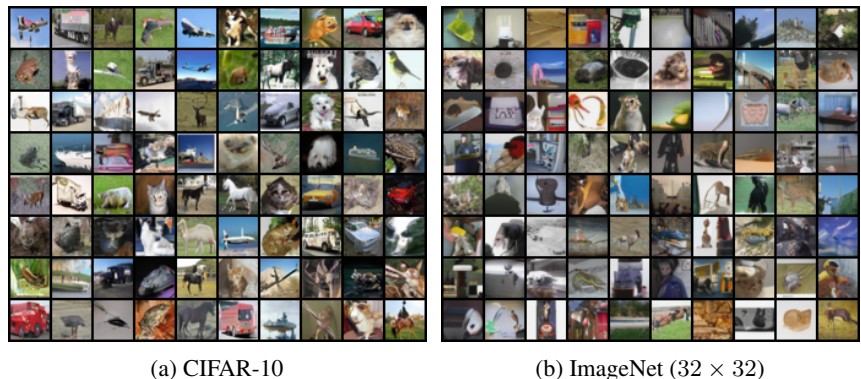

(a) CIFAR-10         (b) ImageNet ($32 \times 32$)

Figure 1: Unconditional generated examples on CIFAR-10 and ImageNet ($32 \times 32$) datasets.

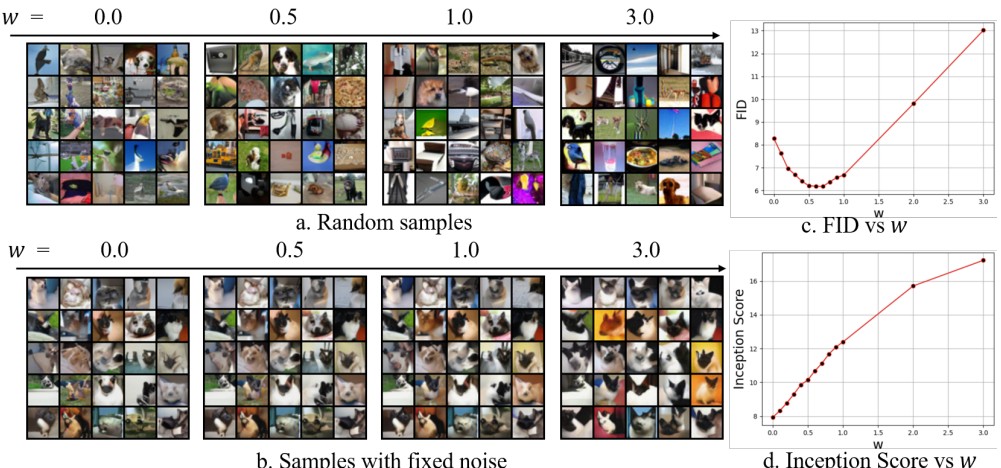

Figure 2: Conditional generation on ImageNet ($32 \times 32$) dataset with a classifier-free guidance. (a) Random image samples generated with different guided weights $w = 0.0, 0.5, 1.0$ and $3.0$; (b) Samples generated with a fixed noise under different guided weights. The class label is set to be the category of Siamese Cat. Sub-images presented at the same position depict samples with identical random noise and class label, differing only in their guided weights; (c) A curve of FID scores across different guided weights; (d) A curve of Inception scores across different guided weights.

## 4 EXPERIMENTS

We evaluate the performance of our model across various scenarios. Specifically, Section 4.1 demonstrates the capacity of unconditional generation. Section 4.2 highlights the potential of our model to further optimize sampling efficiency. The focus shifts to conditional generation and classifier-free guidance in Section 4.3. Section 4.4 elucidates the power of our model in performing likelihood estimation and OOD detection, and Section 4.5 showcases compositional generation. Please refer to Appendix B for implementation details, Appendix C.1 for image inpainting with our trained models, Appendix D.3 for comparing the sampling time between our approach and other EBM

Table 1: Comparison of FID scores for unconditional generation on CIFAR-10.

| Models | FID ↓ | Models | FID ↓ |
|---|---|---|---|
| **EBM based method** | | **Other likelihood based method** | |
| NT-EBM (Nijkamp et al., 2022) | 78.12 | VAE (Kingma & Welling, 2014) | 78.41 |
| LP-EBM (Pang et al., 2020) | 70.15 | PixelCNN (Salimans et al., 2017) | 65.93 |
| Adaptive CE (Xiao & Han, 2022) | 65.01 | PixelIQN (Ostrovski et al., 2018) | 49.46 |
| EBM-SR (Nijkamp et al., 2019) | 44.50 | Residual Flow (Chen et al., 2019) | 47.37 |
| JEM (Grathwohl et al., 2020) | 38.40 | Glow (Kingma & Dhariwal, 2018) | 45.99 |
| EBM-IG (Du & Mordatch, 2019) | 38.20 | DC-VAE (Parmar et al., 2021) | 17.90 |
| EBM-FCE (Gao et al., 2020) | 37.30 | **GAN based method** | |
| CoopVAEBM (Xie et al., 2021b) | 36.20 | | |
| CoopNets (Xie et al., 2018a) | 33.61 | WGAN-GP(Gulrajani et al., 2017) | 36.40 |
| Divergence Triangle (Han et al., 2020) | 30.10 | SN-GAN (Miyato et al., 2018) | 21.70 |
| VARA (Grathwohl et al., 2021) | 27.50 | BigGAN (Brock et al., 2019) | 14.80 |
| EBM-CD (Du et al., 2021) | 25.10 | StyleGAN2-DiffAugment (Zhao et al., 2020) | 5.79 |
| GEBM (Arbel et al., 2021) | 19.31 | Diffusion-GAN (Xiao et al., 2022) | 3.75 |
| HAT-EBM (Hill et al., 2022) | 19.30 | StyleGAN2-ADA (Karras et al., 2020) | 2.92 |
| CF-EBM (Zhao et al., 2021) | 16.71 | **Score based and Diffusion method** | |
| CoopFlow (Xie et al., 2022b) | 15.80 | | |
| CLEL-base (Lee et al., 2023) | 15.27 | NCSN (Song & Ermon, 2019) | 25.32 |
| VAEBM (Xiao et al., 2021) | 12.16 | NCSN-v2 (Song & Ermon, 2020) | 10.87 |
| DRL (Gao et al., 2021) | 9.58 | NCSN++ (Song et al., 2021b) | 2.20 |
| CLEL-large (Lee et al., 2023) | 8.61 | DDPM Distillation (Luhman & Luhman, 2021) | 9.36 |
| EGC (Unsupervised) (Guo et al., 2023) | 5.36 | DDPM++(VP, NLL) (Kim et al., 2021) | 3.45 |
| **CDRL (Ours)** | **4.31** | DDPM (Ho et al., 2020) | 3.17 |
| **CDRL-large (Ours)** | **3.68** | DDPM++(VP, FID) (Kim et al., 2021) | 2.47 |

models, Appendix D.4 for understanding the role of EBM and initializer in the generation process and Appendix D for the ablation study. We designate our approach as "CDRL" in the following sections.

Our experiments primarily involve three datasets: (i) CIFAR-10 (Krizhevsky & Hinton, 2009) comprises images from 10 categories, with 50k training samples and 10k test samples at a resolution of $32 \times 32$ pixels. We use its training set for evaluating our model in the task of unconditional generation. (ii) ImageNet (Deng et al., 2009) contains approximately 1.28M images from 1000 categories. We use its training set for both conditional and unconditional generation, focusing on a downsampled version ($32 \times 32$) of the dataset. (iii) CelebA (Liu et al., 2015) consists of around 200k human face images, each annotated with attributes. We downsample each image of the dataset to the size of $64 \times 64$ pixels and utilize the resized dataset for compositionality and image inpainting tasks.

### 4.1 UNCONDITIONAL IMAGE GENERATION

We first showcase our model's capabilities in unconditional image generation on CIFAR-10 and ImageNet datasets. The resolution of each image is $32 \times 32$ pixels. FID scores (Heusel et al., 2017) on these two datasets are reported in Tables 1 and 3, respectively, with generated examples displayed in Figure 1. We adopt the EBM architecture proposed in Gao et al. (2021). Additionally, we utilize a larger version called "CDRL-large", which incorporates twice as many channels in each layer. For the initializer network, we follow the structure of (Nichol & Dhariwal, 2021), utilizing a U-Net (Ronneberger et al., 2015) but halving the number of channels. Compared to Gao et al. (2021), CDRL achieves significant improvements in FID scores. Furthermore, CDRL uses the same number of noise levels (6 in total) as DRL but requires only half the MCMC steps at each noise level, reducing it from 30 to 15. This substantial reduction in computational costs is noteworthy. With the large architecture, CDRL achieves a FID score of 3.68 on CIFAR-10 and 9.35 on ImageNet ($32 \times 32$). These results, to the best of our knowledge, are state-of-the-art among existing EBM frameworks and are competitive with other strong generative model classes such as GANs and diffusion models.

### 4.2 SAMPLING EFFICIENCY

Similar to the sampling acceleration techniques employed in the diffusion model (Song et al., 2021a; Liu et al., 2022a; Lu et al., 2022), we foresee the development of post-training techniques to further accelerate CDRL sampling. Although designing an advanced MCMC sampling algorithm could

be a standalone project, we present a straightforward yet effective sampling adjustment technique to demonstrate CDRL's potential in further reducing sampling time. Specifically, we propose to decrease the number of sampling steps while simultaneously adjusting the MCMC sampling step size to be inversely proportional to the square root of the number of sampling steps. As shown in Table 2, while we train CDRL with 15 MCMC steps at each noise level, we can reduce the number of MCMC steps to 8, 5, and 3 during the inference stage, without sacrificing much perceptual quality.

### 4.3 CONDITIONAL SYNTHESIS WITH CLASSIFIER-FREE GUIDANCE

We evaluate our model for conditional generation on the ImageNet32 dataset, employing classifier-free guidance as outlined in Section 3.5. Generation results for varying guided weights $w$ are displayed in Figure 2. As the value of $w$ increases, the quality of samples improves, and the conditioned class features become more prominent, although diversity may decrease. This trend is also evident from the FID and Inception Score (Salimans et al., 2016) curves shown in Figures 2(c) and 2(d). While the Inception Score consistently increases (improving quality), the FID metric first drops (improving quality) and then increases (worsening quality), obtaining the optimal value of 6.18 (lowest value) at a guidance weight of 0.7. Additional image generation results can be found in Appendix C.2.

Table 2: FID for CIFAR-10 with sampling adjustment.

| Models | Number of noise level × Number of MCMC steps | FID ↓ |
|---|---|---|
| DRL (Gao et al., 2021) | $6 \times 30 = 180$ | 9.58 |
| CDRL | $6 \times 15 = 90$ | 4.31 |
| CDRL (step 8) | $6 \times 8 = 48$ | 4.58 |
| CDRL (step 5) | $6 \times 5 = 30$ | 5.37 |
| CDRL (step 3) | $6 \times 3 = 18$ | 9.67 |

Table 3: FID for ImageNet ($32 \times 32$) unconditional generation.

| Models | FID ↓ |
|---|---|
| EBM-IG (Du & Mordatch, 2019) | 60.23 |
| PixelCNN (Salimans et al., 2017) | 40.51 |
| EBM-CD (Du et al., 2021) | 32.48 |
| CF-EBM (Zhao et al., 2021) | 26.31 |
| CLEL-base (Lee et al., 2023) | 22.16 |
| DRL (Gao et al., 2021) | - (not converge) |
| DDPM++(VP, NLL) (Kim et al., 2021) | 8.42 |
| **CDRL (Ours)** | 9.35 |

### 4.4 LIKELIHOOD ESTIMATION AND OUT-OF-DISTRIBUTION DETECTION

A distinctive feature of the EBM is its ability to model the unnormalized log-likelihood directly using the energy function. This capability enables it to perform tasks beyond generation. In this section, we first showcase the capability of the CDRL in estimating the density of a 2D checkerboard distribution. Experimental results are presented in Figure 3, where we illustrate observed samples, the fitted density, and the generated samples at each noise level, respectively. These results confirm CDRL's ability to accurately estimate log-likelihood while simultaneously generating valid samples.

Moreover, we demonstrate CDRL's utility in out-of-distribution (OOD) detection tasks. For this endeavor, we employ the model trained on CIFAR-10 as a detector and use the energy at the lowest

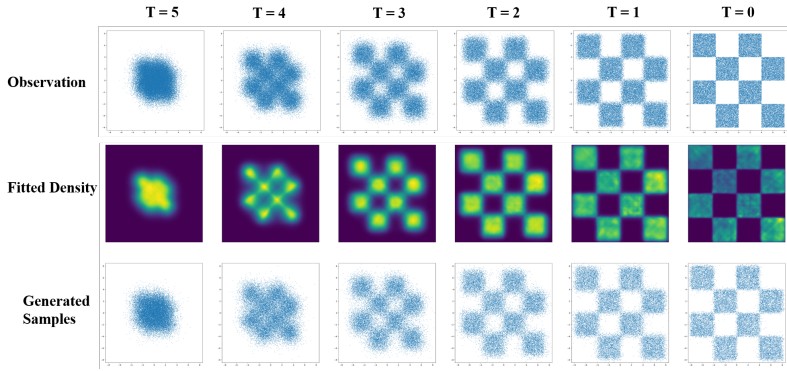

Figure 3: The results of density estimation using CDRL for a 2D checkerboard distribution. The number of noise levels in the CDRL is set to be 5. Top: observed samples at each noise level. Middle: density fitted by CDRL at each noise level. Bottom: generated samples at each noise level.

noise level to serve as the OOD prediction score. The AUROC score of our CDRL model, with CIFAR-10 interpolation, CIFAR-100, and CelebA data as OOD samples, is provided in Table 4. CDRL achieves strong results in OOD detection comparing with the baseline approaches. More results can be found in Table 8 in the appendix.

## 4.5 COMPOSITIONALITY

To evaluate the compositionality of EBMs, we conduct experiments on CelebA ($64 \times 64$) datasets with *Male*, *Smile*, and *Young* as the three conditional concepts. We estimate EBMs conditional on each single concept separately, and assume simple unconditional initializer models. Classifier-free guidance is adopted when conducting compositional generation (Equation 14). Specifically, we treat images with a certain attribute value as individual classes. We randomly assign each image in a training

Table 4: AUROC scores in OOD detection using CDRL and other explicit density models on CIFAR-10

|  | Cifar-10 interpolation | Cifar-100 | CelebA |
|---|---|---|---|
| PixelCNN (Salimans et al., 2017) | 0.71 | 0.63 | - |
| GLOW (Kingma & Dhariwal, 2018) | 0.51 | 0.55 | 0.57 |
| NVAE (Vahdat & Kautz, 2020) | 0.64 | 0.56 | 0.68 |
| EBM-IG (Du & Mordatch, 2019) | 0.70 | 0.50 | 0.70 |
| VAEBM (Xiao et al., 2021) | 0.70 | 0.62 | 0.77 |
| EBM-CD (Du et al., 2021) | 0.65 | 0.83 | - |
| CLEL-Base (Lee et al., 2023) | 0.72 | 0.72 | 0.77 |
| **CDRL (ours)** | 0.75 | 0.78 | 0.84 |

batch to a class based on the controlled attribute value. For example, an image with Male=True and Smile=True may be assigned to class 0 if the Male attribute is picked or class 2 if the Smile attribute is picked. For the conditional network structure, we make EBM $f_\theta$ conditional on attributes $c_i$ and use an unconditional initializer model $g_\phi$ to propose the initial distribution. We focus on showcasing the compositionality ability of EBM itself, although it is also possible to use a conditional initializer model similar to Section 3.5. Our results are displayed in Figure 4, with images generated at a guided weight of $w = 3.0$. More generation results with different guidance weights can be found in the Appendix C.2. Images generated with composed attributes following Equation 14 contain features of both attributes, and increasing the guided weight makes the corresponding attribute more prominent. This demonstrates CDRL's ability and the effectiveness of Equation 14.

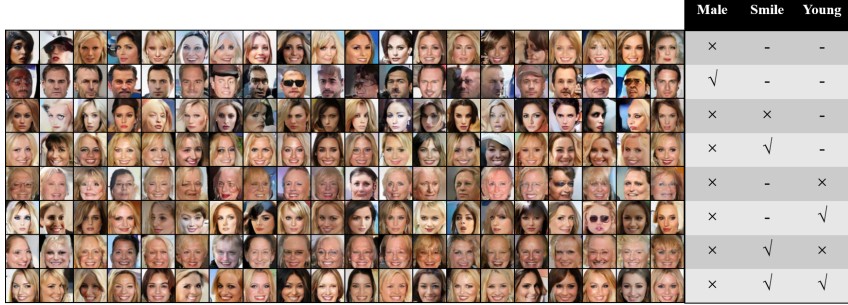

Figure 4: Results of attribute-compositional generation on CelebA ($64 \times 64$) with guided weight $w = 3$. Left: generated samples under different attribute compositions. Right: control attributes ("$\sqrt{}$", "$\times$" and "-" indicate "True", "False" and "No Control" respectively).

## 5 CONCLUSION AND FUTURE WORK

We propose CDRL, a novel energy-based generative learning framework employing cooperative diffusion recovery likelihood, which significantly enhances the generation performance of EBMs. We demonstrate that the CDRL excels in compositional generation, out-of-distribution detection, image inpainting, and compatibility with classifier-free guidance for conditional generation. One limitation is that a certain number of MCMC steps are still needed during generation. Additionally, we aim to scale our model for high-resolution image generation in the future. Our work aims to stimulate further research on developing EBMs as generative models. However, the prevalence of powerful generative models may give rise to negative social consequences, such as deepfakes, misinformation, privacy breaches, and erosion of public trust, highlighting the need for effective preventive measures.

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

## APPENDICES

## A  RELATED WORK

**Energy-Based Learning** Energy-based models (EBMs) (Zhu et al., 1998; LeCun et al., 2006; Ngiam et al., 2011; Hinton, 2012; Xie et al., 2016) define unnormalized probabilistic distributions and are typically trained through maximum likelihood estimation. Methods such as contrastive divergence (Hinton, 2002; Du et al., 2021), persistent chain (Xie et al., 2016), replay buffer (Du & Mordatch, 2019) or short-run MCMC sampling (Nijkamp et al., 2019) approximate the analytically intractable learning gradient. To scale up and stabilize EBM training for high-fidelity data generation, strategies like multi-grid sampling (Gao et al., 2018), progressive training (Zhao et al., 2021), and diffusion (Gao et al., 2021) have been adopted. EBMs have also been connected to other models, such as adversarial training (Arbel et al., 2021; Che et al., 2020), variational autoencoders (Xiao et al., 2021), contrastive guidance (Lee et al., 2023), introspective learning (Lazarow et al., 2017; Jin et al., 2017; Lee et al., 2018), and noise contrastive estimation (Gao et al., 2020). To alleviate MCMC burden, various methods have been proposed, including amortizing MCMC sampling with learned networks (Kim & Bengio, 2016; Xie et al., 2018a; Kumar et al., 2019; Xiao et al., 2021; Han et al., 2019; Grathwohl et al., 2021). Among them, cooperative networks (CoopNets) (Xie et al., 2018a) jointly train a top-down generator and an EBM via MCMC teaching, using the generator as a fast initializer for Langevin sampling. CoopNets variants have also been studied in Xie et al. (2021b; 2022b). Our work improves the recovery likelihood learning algorithm of EBMs (Gao et al., 2021) by learning a fast MCMC initializer for EBM sampling, leveraging the cooperative learning scheme (Xie et al., 2020). Compared to Xie et al. (2020) that applied cooperative training to an initializer and an EBM for the *marginal* distribution of the clean data, our approach only requires learning *conditional* initializers and sampling from *conditional* EBMs, which are much more tractable than their marginal counterparts. It is worth noting that Xie et al. (2022a) have investigated conditional cooperative learning, wherein a conditional initializer and a conditional EBM are trained through MCMC teaching. In contrast, our CDRL trains and samples from a sequence of conditional EBMs and

conditional initializers on increasingly noisy versions of a dataset for denoising diffusion generation.

**Denoising Diffusion Model** Diffusion models, initially introduced by Sohl-Dickstein et al. (2015) and further developed in subsequent works such as Song & Ermon (2020); Ho et al. (2020), generate samples by progressively denoising them from a high noise level to clean data. These models have demonstrated significant success in generating high-quality samples from complex distributions, owing to a range of architectural and algorithmic innovations (Ho et al., 2020; Song et al., 2021a; Kim et al., 2021; Song et al., 2021b; Dhariwal & Nichol, 2021; Karras et al., 2022; Ho & Salimans, 2022). Notably, Dhariwal & Nichol (2021) emphasize that the generative performance of diffusion models can be enhanced with the aid of a classifier, while Ho & Salimans (2022) further demonstrate that this guided scoring can be estimated by the differential scores of a conditional model versus an unconditional model. Enhancements in sampling speed have been realized through distillation techniques (Salimans & Ho, 2022) and the development of fast SDE/ODE samplers (Song et al., 2021a; Karras et al., 2022; Lu et al., 2022). Recent advancements (Rombach et al., 2022; Saharia et al., 2022; Ramesh et al., 2022) have successfully applied conditional diffusion models to the task of text-to-image generation, achieving significant breakthroughs.

EBM shares a close relationship with diffusion models, as both frameworks can provide a score to guide the generation process, whether through Langevin dynamics or SDE/ODE solvers. As Salimans & Ho (2021) discuss, the distinction between these two models lies in their implementation approaches: EBMs model the log-likelihood directly, while diffusion models focus on the gradient of the log-likelihood. This distinction brings advantages to EBMs, such as their compatibility with advanced sampling techniques (Du et al., 2023), potential conversion into classifiers (Guo et al., 2023), and capability to detect abnormal samples through estimated likelihood (Grathwohl et al., 2020; Liu et al., 2020).

The primary focus of this work is to advance the development of EBMs. Our approach connects with diffusion models (Ho et al., 2020; Xiao et al., 2022) by training a sequence of EBMs and MCMC initializers to reverse the diffusion process. In contrast to Ho et al. (2020), our framework employs more expressive conditional EBMs instead of normal distributions to represent the denoising distribution. Additionally, Xiao et al. (2022) also suggest multimodal distributions, trained by generative adversarial networks (Goodfellow et al., 2020), for the reverse process.

## B  TRAINING DETAILS

### B.1  NETWORK ARCHITECTURES

We adopt the EBM architecture from (Gao et al., 2021), starting with a $3 \times 3$ convolution layer with 128 channels (The number of channels is doubled to 256 in the CDRL-large configuration). We use several downsample blocks for resolution adjustments, each containing multiple residual blocks. All downsampling blocks, except the last one, include a $2 \times 2$ average pooling layer. Spectral normalization is applied to all convolution layers for stability, while ReLU activation is applied to the final feature map. The energy output is obtained by summing the values over spatial and channel dimensions. The architectures of EBM building blocks are shown in Table 5, and the hyperparameters of network architecture are displayed in Table 6.

For the initializer network, we follow (Nichol & Dhariwal, 2021) to utilize a U-Net (Ronneberger et al., 2015) while halving the number of channels. This reduction effectively decreases the size of the initializer model. For an image with a resolution of $32 \times 32$ pixels, we have feature map resolutions of $32 \times 32$, $16 \times 16$, and $4 \times 4$. When dealing with $64 \times 64$ images, we include an additional feature map resolution of $64 \times 64$. All feature map channel numbers are set to 64, with attention applied to resolutions of $16 \times 16$ and $8 \times 8$. Our initializer directly predicts the noised image $\tilde{y}_t$ at each noise level $t$, while the DDPM in (Ho et al., 2020) predicts the total injected noise $\epsilon$.

For the class-conditioned generation task, we map class labels to one-hot vectors and use a fully-connected layer to map these vectors to class embedding vectors with the same dimensions as time embedding vectors. The class embedding is then added to the time embedding. We set the time embedding dimension to 512 for EBM and 256 for the initializer in the CDRL setting. In the

CDRL-large setting, the time embedding dimension increases to 1024 for EBM, while the one in the initializer remains unchanged.

Table 5: Building blocks of the EBM in CDRL.

| (a) ResBlock |
| --- |
| leakyReLU, $3 \times 3$ Conv2D |
| + Dense(leakyReLU(temb)) |
| leakyReLU, $3 \times 3$ Conv2D |
| + Input |

| (b) Downsample Block |
| --- |
| $N$ ResBlocks |
| Downsample $2 \times 2$ |

| (c) Time Embedding |
| --- |
| Sinusoidal Embedding |
| Dense, leakyReLU |
| Dense |

Table 6: Hyperparameters for EBM architectures in different settings.

| Model | # of Downsample Blocks | $N$ (# of Resblocks in Downsample Block) | # of channels in each resolution |
| --- | --- | --- | --- |
| CDRL ($32 \times 32$) | 4 | 8 | (128, 256, 256, 256) |
| CDRL-large ($32 \times 32$) | 4 | 8 | (256, 512, 512, 512) |
| Compositionality Exp. | 5 | 2 | (128, 256, 256, 256, 256) |
| Inpainting Exp. | 5 | 8 | (128, 256, 256, 256, 256) |

## B.2 HYPERPARAMETERS

We set the learning rate of EBM to be $\eta_\theta = 1e-4$ and the learning rate of initializer to be $\eta_\phi = 1e-5$. We use linear warm up for both EBM and initializer and let the initializer to start earlier than EBM. More specifically, given training iteration iter, we have:

$$\eta_\theta = \min(1.0, \frac{\text{iter}}{10000}) \times 1e - 4$$
$$\eta_\phi = \min(1.0, \frac{\text{iter} + 500}{10000}) \times 1e - 5$$

(15)

We use the Adam optimizer Kingma & Ba (2015); Loshchilov & Hutter (2019) to train both the EBM and the initializer, with $\beta = (0.9, 0.999)$ and a weight decay equal to 0.0. We also apply exponential moving average with a decay rate equal to 0.9999 to both the EBM and the initializer. Training is conducted across 8 Nvidia A100 GPUs, typically requiring approximately 400k iterations, which spans approximately 6 days.

Following (Gao et al., 2021), we use a re-parameterization trick to calculate the energy term. Our EBM is constructed across noise levels $t = 0, 1, 2, 3, 4, 5$ and we assume the distribution at noise level $t = 6$ is a simple Normal distribution during sampling. Given $\mathbf{y}_t$ under noise level $t$, suppose we denote the output of the EBM network as $\hat{f}_\theta(\mathbf{y}_t, t)$, then the true energy term is given by $f_\theta(\mathbf{y}_t, t) = \frac{\hat{f}_\theta(\mathbf{y}_t, t)}{s_t^2}$, where $s_t$ is the Langevin step size at noise level $t$. In other words, we parameterize the energy as the product of the EBM network output and a noise-level dependent coefficient, setting this coefficient equal to the square of the Langevin step size. We use 15 steps of Langevin updates at each noise level, with the Langevin step size at noise level $t$ given by

$$s_t^2 = 0.054 \times \bar{\sigma}_t \times \sigma_{t+1}^2,$$

(16)

where $\sigma_{t+1}^2$ is the variance of the added noise at noise level $t+1$ and $\bar{\sigma}_t$ is the standard deviation of the accumulative noise at noise level $t$. During the generation process, we begin by randomly sampling $\mathbf{x}_6 \sim \mathcal{N}(0, \mathbf{I})$ and perform denoising using both the initializer and the Langevin Dynamics of the EBM, which follows Algorithm 2. After obtaining samples $\mathbf{x}_0$ at the lowest noise level $t = 0$, we perform an additional denoising step, where we disable the noise term in the Langevin step, to further enhance its quality. More specifically, we follow Tweedie's formula (Efron, 2011; Robbins, 1992),

which states that given $\mathbf{x} \sim p_{data}(\mathbf{x})$ and a noisy version image $\mathbf{x}'$ with conditional distribution $p(\mathbf{x}'|\mathbf{x}) = \mathcal{N}(\mathbf{x}, \sigma^2 \boldsymbol{I})$, the marginal distribution can be defined as $p(\mathbf{x}') = \int p_{data}(\mathbf{x})p(\mathbf{x}'|\mathbf{x})d\mathbf{x}$. Consequently, we have

$$\mathbb{E}(\mathbf{x}|\mathbf{x}') = \mathbf{x}' + \sigma^2 \nabla_{\mathbf{x}'} \log p(\mathbf{x}'). \tag{17}$$

In our case, we have $p(\mathbf{x}_t|\bar{\alpha}_t \mathbf{x}_0) = \mathcal{N}(\bar{\alpha}_t \mathbf{x}_0, \bar{\sigma}_t^2 \boldsymbol{I})$ and we use EBM to model the marginal distribution of $\mathbf{x}_t$ as $p_{\theta,t}(\mathbf{x}_t)$, thus

$$\mathbb{E}(\bar{\alpha}_t \mathbf{x}_0|\mathbf{x}_t) = \mathbf{x}_t + \bar{\sigma}_t^2 \nabla_{\mathbf{x}_t} \log p_{\theta,t}(\mathbf{x}_t),$$
$$\mathbb{E}(\mathbf{x}_0|\mathbf{x}_t) = \frac{\mathbf{x}_t + \bar{\sigma}_t^2 \nabla_{\mathbf{x}_t} \log p_{\theta,t}(\mathbf{x}_t)}{\bar{\alpha}_t}. \tag{18}$$

Suppose the samples we obtain at $t = 0$ are denoted as $\mathbf{x}_0$. These samples actually contains a small amount of noise corresponding to $\bar{\alpha}_0$, thus, we may use Equation 18 to further denoise them. In practice, we find that enlarging the denoising step by multiplying the gradient term $\nabla_{\mathbf{x}_t} \log p_{\theta,t}(\mathbf{x}_t)$ by a coefficient larger than 1.0 yields better results. We set this coefficient to be 2.0 in our experiments.

### B.3 NOISE SCHEDULE AND CONDITIONING INPUT

We improve upon the noise schedule and the conditioning input of DRL (Gao et al., 2021). Let $\lambda_t = \log \frac{\bar{\alpha}_t^2}{\bar{\sigma}_t^2}$ represent the logarithm of signal-to-noise ratio at noise level $t$. Inspired by (Kingma et al., 2021), we utilize $\lambda_t$ as the conditioning input of the noise level and feed it to the networks $f_\theta$ and $\boldsymbol{g}_\phi$ instead of directly using $t$.

For the noise schedule, we keep the design of using 6 noise levels as in DRL. Inspired by (Nichol & Dhariwal, 2021), we construct a cosine schedule such that $\lambda_t$ is defined as $\lambda_t = -2\log(\tan(at + b))$, where $a$ and $b$ are calculated from the maximum log SNR (denoted as $\lambda_{\max}$) and the minimum log SNR (denoted as $\lambda_{\min}$) using

$$b = \arctan(\exp(-0.5\lambda_{\max})), \tag{19}$$
$$a = \arctan(\exp(-0.5\lambda_{\min})) - b. \tag{20}$$

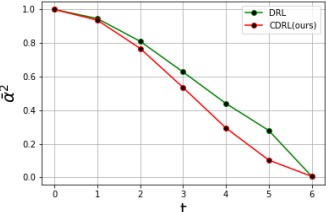

We set $\lambda_{\max} = 9.8$ and $\lambda_{\min} = -5.1$ to correspond with the standard deviation $\bar{\alpha}_t$ of the accumulative noise in the original Recovery Likelihood model (T6 setting) at the highest and lowest noise levels. Figure 5 illustrates the noise schedule of DRL alongside our proposed schedule. In contrast to the DRL's original schedule, our proposed schedule places more emphasis on regions with lower signal-to-noise ratios, which are vital for generating low-frequency, high-level concepts in samples.

Figure 5: Noise schedule. The green line represents the noise schedule used by DRL (Gao et al., 2021) while the red line depicts the noise schedule employed by our CDRL.

### B.4 ILLUSTRATION OF THE CDRL FRAMEWORK

Figure 6 illustrates the training and sampling processes of the CDRL framework, providing a comprehensive overview of the model.

## C MORE EXPERIMENTAL RESULTS

### C.1 IMAGE INPAINTING

We demonstrate the inpainting ability of our learned model on the $64 \times 64$ CelebA dataset. Each image is masked, and our model is tasked with filling in the masked area. We gradually add noise to the masked image up to the final noise level, allowing the model to denoise the image progressively, similar to the standard generation process. During inpainting, only the masked area is updated, while the values in the unmasked area are retained. This is achieved by resetting the unmasked area values to the current noisy version after each Langevin update step of the EBM or initializer proposal step. Our results, depicted in Figure 7, include two types of masking: a regular square mask and an

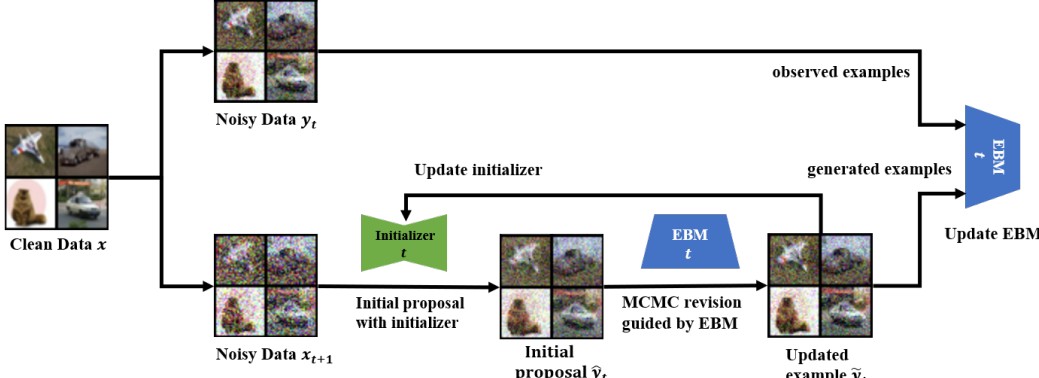

(a) CDRL training process. In the training phase, we start by selecting a pair of images $\{\mathbf{y}_t, \mathbf{x}_{t+1}\}$ at noise levels $t$ and $t + 1$ respectively. The image $\mathbf{x}_t$ at noise level $t$ is then fed into the initializer to generate an initial proposal $\hat{\mathbf{y}}_t$. Subsequently, this initial proposal undergoes refinement through the MCMC process guided by the underlying energy function. The refined sample $\tilde{\mathbf{y}}_t$ obtained from this process is utilized to update both the energy function and the initializer.

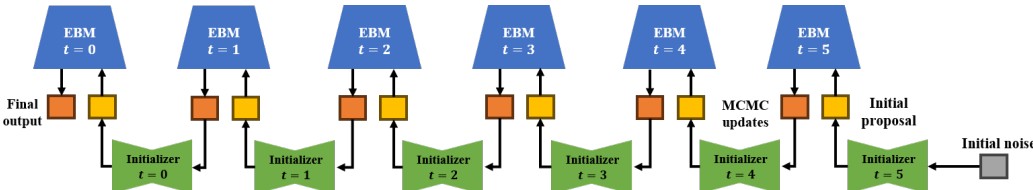

(b) CDRL Sampling process. The sampling phase starts from Gaussian noise. Starting from the highest noise level, an initial proposal is generated by the initializer that corresponds to that noise level. Subsequently, the samples undergo refinement through MCMC sampling. This denoising process is iteratively repeated to push the noisy image towards lower noise levels until the lowest noise level $t = 0$ is reached.

Figure 6: Illustration of the Cooperative Diffusion Recovery Likelihood (CDRL) framework

irregularly shaped mask. In Figure 7, the first two columns respectively display the original images and the masked images, while the other columns show the corresponding inpainting results. CDRL successfully inpaints valid and diverse values in the masked area, producing inpainted results that differ from the observations. This suggest that CDRL does not merely memorize data because it fills novel and meaningful content into unobserved areas based on the statistical features of the dataset.

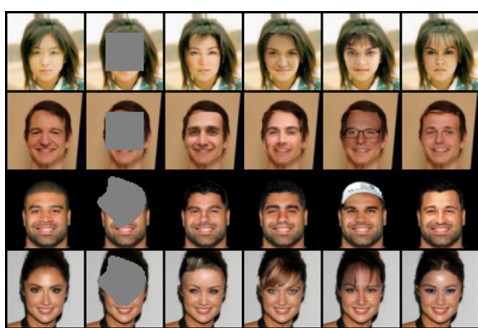

Figure 7: Results of Image inpainting on CelebA ($64 \times 64$) dataset. The first two rows utilize square masks, while the last two rows use irregular masks. The first column displays the original images. The second column shows the masked images. Columns three to six display inpainted images using different initialization noises.

## C.2 IMAGE GENERATION

In this section we present additional generation results. Figure 10 showcases more compositionality results with varying guidance weights on the CelebA $64 \times 64$ dataset. Here, $w = 0.0$ corresponds to the original setting without guidance in Equation 13 in the main paper. In Figure 11, 12, 13, 14 and 15, we provide more results for conditional generation on ImageNet32 ($32 \times 32$) with different guidance weights. Specifically, Figure 11 showcases random samples, while each figure in Figures 12, 13, 14 and 15 contains samples from a specific class under different guidance weights $w$.

## C.3 GENERATING HIGH-RESOLUTION IMAGES

The recent trend in generative modeling of high-resolution images involves either utilizing the latent space of a VAE, as demonstrated in latent diffusion (Rombach et al., 2022), or initially generating a low-resolution image and then gradually expanding it, as exemplified by techniques like Imagen (Saharia et al., 2022). This process often reduce the modeled space to dimensions such as $32 \times 32$ or $64 \times 64$, which aligns with the resolutions that we used in our experiments in the main text. Here, we conduct additional experiments by learning CDRL following Rombach et al. (2022). We conduct experiments on the CelebA-HQ dataset, and the generated samples are shown in Figure 8. Additionally, we report the FIDs in Table 7.

Table 7: Comparison of FIDs on the CelebA-HQ (256 x 256) dataset

| Model | FID score |
|---|---|
| GLOW (Kingma & Dhariwal, 2018) | 68.93 |
| VAEBM (Xiao et al., 2021) | 20.38 |
| ATEBM (Yin et al., 2022) | 17.31 |
| VQGAN+Transformer (Esser et al., 2021) | 10.2 |
| LDM (Rombach et al., 2022) | 5.11 |
| CDRL(ours) | 10.74 |

## C.4 OUT-OF-DISTRIBUTION DETECTION

We present the results for the out-of-distribution (OOD) detection task on additional datasets, along with incorporating more recent baselines. The comprehensive results are summarized in Table 8.

Table 8: Comparison of AUROC scores in OOD detection on CIFAR-10 dataset. The AUROC score for DRL (Gao et al., 2021) is reported by Yoon et al. (2023). For EBM-CD (Du et al., 2021), we present two sets of performance results from different sources: one from Du et al. (2021) and the other from a recent study (Yoon et al., 2023). We include both sets of results, with scores from Yoon et al. (2023) presented in brackets for comparison.

| | Cifar-10 interpolation | Cifar-100 | CelebA | SVHN | Texture |
|---|---|---|---|---|---|
| **PixelCNN**(Salimans et al., 2017) | 0.71 | 0.63 | - | 0.32 | 0.33 |
| **GLOW** (Kingma & Dhariwal, 2018) | 0.51 | 0.55 | 0.57 | 0.24 | 0.27 |
| **NVAE** (Vahdat & Kautz, 2020) | 0.64 | 0.56 | 0.68 | 0.42 | - |
| **EBM-IG** (Du & Mordatch, 2019) | 0.70 | 0.50 | 0.70 | 0.63 | 0.48 |
| **VAEBM** (Xiao et al., 2021) | 0.70 | 0.62 | 0.77 | 0.83 | - |
| **EBM-CD** (Du et al., 2021) | 0.65 (-) | 0.83 (0.53) | - (0.54) | 0.91 (0.78) | 0.88 (0.73) |
| **CLEL** (Lee et al., 2023) | 0.72 | 0.72 | 0.77 | 0.98 | 0.94 |
| **DRL** (Gao et al., 2021) | - | 0.44 | 0.64 | 0.88 | 0.45 |
| **MPDR-S**(Yoon et al., 2023) | - | 0.56 | 0.73 | 0.99 | 0.66 |
| **MPDR_R**(Yoon et al., 2023) | - | 0.64 | 0.83 | 0.98 | 0.80 |
| **CDRL** | 0.75 | 0.78 | 0.84 | 0.82 | 0.65 |

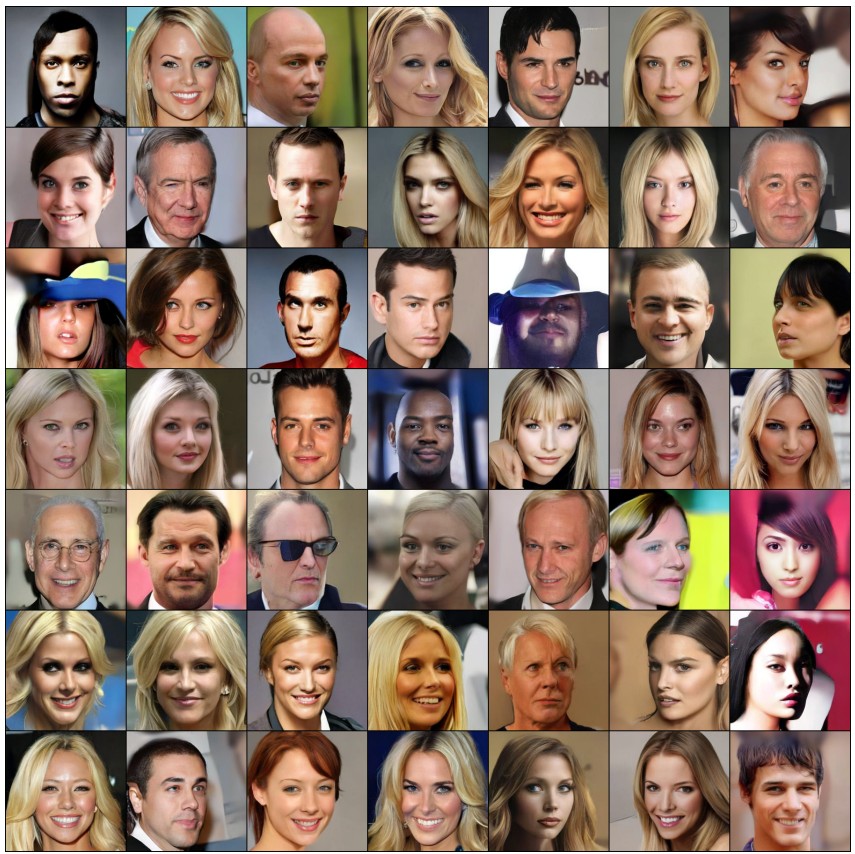

Figure 8: Samples generated by CDRL model trained on the CelebAHQ ($256 \times 256$) dataset.

# D    MODEL ANALYSIS

In this section, we employ several experiments to analyze the CDRL model.

## D.1    ABLATION STUDY

In this section, we conduct an ablation study to analyze the effectiveness of each component of our CDRL model. We have previously described three main techniques in our main paper that contribute significantly to our CDRL model: (1) the new noise schedule design, (2) the cooperative training algorithm, and (3) noise variance reduction. We demonstrate the impact of each of these techniques by comparing our CDRL model with the following models:

1. The original diffusion recovery likelihood (DRL) model (Gao et al., 2021) as a baseline.

2. A model trained without using the cooperative training. This corresponds to the DRL but using the same noise schedule and conditioning input as CDRL.

3. CDRL without using noise reduction.

4. Similar to Xiao et al. (2022), we use the initializer to predict the clean image $\hat{\mathbf{x}}_0$ and then transform it to $\hat{\mathbf{y}}_t$. Note that our CDRL uses the initializer to directly predict $\hat{\mathbf{y}}_t$.

5. Similar to Ho et al. (2020), we use initializer to directly output the prediction of total added noise $\hat{\epsilon}$ and then transform it to $\hat{\mathbf{y}}_t$.

6. Compared with the noise schedule used in the original DRL(Gao et al., 2021) paper, the proposed one used in our CDRL places more emphasis on the high-noise area where $\bar{\alpha}$ is close to 0. We train a CDRL model with the original DRL noise schedule but with 2 additional noise levels in the high-noise region for comparison.

7. As depicted in Equation 8 in the main paper, our cooperative training algorithm involves the initializer learning from the revised sample $\tilde{\mathbf{y}}_t$ at each step. A natural question arises: should we instead regress it directly on the data $\mathbf{y}_t$? To answer this, we train a model, in which the initializer directly learns from $\mathbf{y}_t$ at each step.

We ensure that all models share the same network structure and training settings on the CIFAR-10 dataset and differ only in the aforementioned ways. As shown in Table 9, our full model performs the best among these settings, which justifies our design choices.

Table 9: Ablation study on the CIFAR-10 dataset.

| Models | FID ↓ |
|---|---|
| DRL (Gao et al., 2021) | 9.58 |
| CDRL without cooperative training | 6.47 |
| CDRL without noise reduction | 5.51 |
| CDRL with an initializer that predicts $\hat{\mathbf{x}}_0$ | 5.17 |
| CDRL with an initializer that predicts $\hat{\epsilon}$ | 4.95 |
| CDRL using a noise schedule in DRL-T8 | 4.94 |
| CDRL with an initializer that learns from $\mathbf{y}_t$ | 5.95 |
| CDRL (full) | 4.31 |

## D.2 EFFECTS OF NUMBER OF NOISE LEVELS AND NUMBER OF LANGEVIN STEPS

We test whether the noise level can be further reduced. The results in Table 10a show that further reducing noise level to 4 can make model more unstable, even if we increase the number of the Langevin sample steps $K$. On the other hand, reducing T to 5 yields reasonable but slightly worse results. In Table 10b, we show the effect of changing the number of Langevin steps $K$. The results show that, on one hand, decreasing $K$ to 10 yields comparable but slightly worse results. On the other hand, increasing $K$ to 30 doesn't lead to better results. This observation aligns wit the finding from Gao et al. (2021). The observation of changing $K$ implies that simply increasing the number of Langevin steps doesn't significantly enhance sample quality, thereby verifying the effectiveness of the initializer in our model.

Table 10: Comparison of CDRL models with varying numbers of noise levels $T$ and varying numbers of Langevin steps $K$. FIDs are reported on the Cifar-10 dataset.

(a) Results of CDRL models with varying $T$

| Model | FID ↓ |
|---|---|
| $T = 4$ ($K = 15, 20, 30$) | not converge |
| $T = 5$ ($K = 15$) | 5.08 |
| $T = 6$ ($K = 15$) | 4.31 |

(b) Results of CDRL models with varying $K$

| Model | FID ↓ |
|---|---|
| $T = 6$ ($K = 10$) | 4.50 |
| $T = 6$ ($K = 15$) | 4.31 |
| $T = 6$ ($K = 30$) | 5.08 |

## D.3 SAMPLING TIME

In this section, we measure the sampling time of CDRL and compare it with the following models: (1) CoopFlow (Xie et al., 2022b), which combines an EBM with a Normalizing Flow model; (2) VAEBM (Xiao et al., 2021), which combines a VAE with an EBM and achieves strong generation performance; (3) DRL (Gao et al., 2021) model with a 30-step MCMC sampling at each noise level. We conduct the sampling process of each model individually on a single A6000 GPU to generate a batch of 100 samples on the Cifar10 dataset. Our CDRL model produces samples with better quality with a relatively shorter time frame. Additionally, with the sampling adjustment techniques, the sampling time can be further reduced without significantly compromising sampling quality.

Table 11: Comparison of different EBMs in terms of sampling time and number of MCMC steps. The sampling time are measured in seconds.

| Method | Number of MCMC steps | Sampling Time | FID $\downarrow$ |
|---|---|---|---|
| CoopFlow (Xie et al., 2022b) | 30 | 2.5 | 15.80 |
| VAEBM (Xiao et al., 2021) | 16 | 21.3 | 12.16 |
| DRL (Gao et al., 2021) | $6 \times 30 = 180$ | 23.9 | 9.58 |
| CDRL | $6 \times 15 = 90$ | 12.2 | 4.31 |
| CDRL (8 steps) | $6 \times 8 = 48$ | 6.5 | 4.58 |
| CDRL (5 steps) | $6 \times 5 = 30$ | 4.2 | 5.37 |
| CDRL (3 steps) | $6 \times 3 = 18$ | 2.6 | 9.67 |

### D.4 ANALYZING THE EFFECTS OF THE INITIALIZER AND THE EBM

To gain deeper insights into the roles of the initializer and the EBM in the CDRL in image generation, we conduct two additional experiments using a pretrained CDRL model on the ImageNet Dataset ($32 \times 32$). We evaluate two generation options: (a) images generated using only the initializer's proposal, without the EBM's Langevin Dynamics at each noise level, and (b) images generated with the full CDRL model, which includes the initializer's proposal and 15-step Langevin updates at each noise level. As shown in Figure 9a and 9b, the initializer captures the rough outline of the object, while the Langevin updates by the EBM improve the details of the object. Furthermore, in Figure 9c, we display samples generated by fixing the initial noise image and sample noise of each initializer proposal step. The outcomes demonstrate that images generated with the same initialization noises share basic elements but differ in details, highlighting the impact of both the initializer and the Langevin sampling. The initializer provides a starting point, while the Langevin sampling process enriches details.

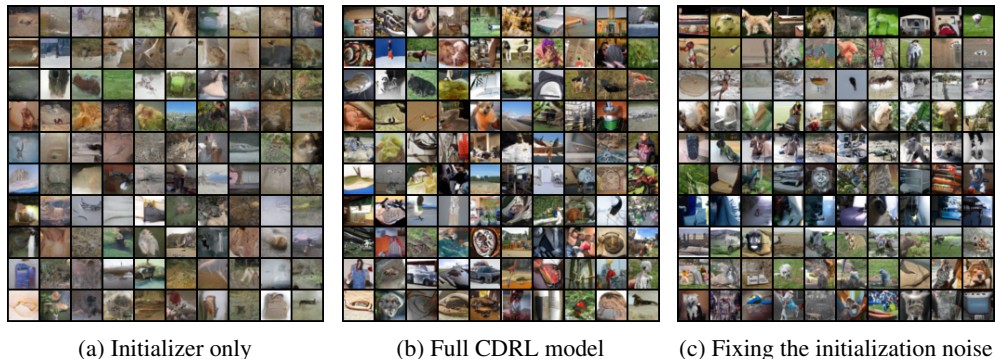

(a) Initializer only  (b) Full CDRL model  (c) Fixing the initialization noise

Figure 9: Illustration of the effects of the initializer and the EBM on the image generation process using a CDRL model pretrained on the ImageNet Dataset ($32 \times 32$). (a) Samples generated using only the proposal of the initializer; (b) Samples generated by the full CDRL model; (c) Samples generated by fixing the initial noise image and the sample noise of each initialization proposal step. Each row of images shared the same initial noise image and the sample noise of each initialization proposal step, but differed in the noises of Langevin sampling process at each noise level.

### D.5 ANALYZING LEARNING BEHAVIOR

We dive into a deeper understanding of the learning behavior of the cooperative learning algorithm. We follow the analysis framework of Xie et al. (2020; 2022a). Let $K_\theta(\mathbf{y}_t|\mathbf{y}_t', \mathbf{x}_{t+1})$ be the transition kernel of the $K$-step Langevin sampling that refines the initial output $\mathbf{y}_t'$ to the refined output $\mathbf{y}_t$. Let $(K_\theta q_\phi)(\mathbf{y}_t|\mathbf{x}_{t+1}) = \int K_\theta(\mathbf{y}_t|\mathbf{y}_t', \mathbf{x}_{t+1})q_\phi(\mathbf{y}_t'|\mathbf{x}_{t+1})d\mathbf{y}_t'$ be the conditional disribution of $\mathbf{y}_t$, which is obtained by $K$ steps of Langevin sampling starting from the output of the initialier $q_\phi(\mathbf{y}_t|\mathbf{x}_{t+1})$. Let $\pi(\mathbf{y}_t|\mathbf{x}_{t+1})$ be the true conditional distribution for denoising $\mathbf{x}_{t+1}$ to retrieve $\mathbf{y}_t$. The maximum recovery likelihood for the EBM in Equation 4 in the main paper is equivalent to minimizing the

Kullback-Leibler divergence (KL) divergence $\text{KL}(\pi(\mathbf{x}_t|\mathbf{y}_{t+1})||p_\theta(\mathbf{x}_t|\mathbf{y}_{t+1}))$. Using $j$ to index the learning iteration for model parameters, given the current initializer model $q_\phi(\mathbf{y}_t|\mathbf{x}_{t+1})$, the EBM updates its parameters $\theta$ by minimizing

$$\theta_{j+1} = \arg\min_\theta \text{KL}(\pi(\mathbf{y}_t|\mathbf{x}_{t+1})||p_\theta(\mathbf{y}_t|\mathbf{x}_{t+1})) - \text{KL}((K_{\theta_j}q_\phi)(\mathbf{y}_t|\mathbf{x}_{t+1})||p_\theta(\mathbf{y}_t|\mathbf{x}_{t+1})) \quad (21)$$

which is a modified contrastive divergence. It is worth noting that, in the original contrastive divergence, the above $(K_{\theta_j}q_\phi)(\mathbf{y}_t|\mathbf{x}_{t+1})$ is replaced by $(K_{\theta_j}\pi)(\mathbf{y}_t|\mathbf{x}_{t+1})$. That is, the MCMC chains are initialized by the true data. The learning shifts $p_\theta(\mathbf{y}_t|\mathbf{x}_{t+1})$ toward the true distribution $\pi(\mathbf{y}_t|\mathbf{x}_{t+1})$.

On the other hand, given the current EBM, the initializer model learns from the output distribution of the EBM's MCMC. That is, we train the initializer with $\tilde{\mathbf{y}}$ in equation 8 of the main paper. The update of the parameters of the initializer at learning iteration $j + 1$ approximately follows the gradient of

$$\phi_{j+1} = \arg\min_\phi \text{KL}(K_\theta q_{\phi_j}(\mathbf{y}_t|\mathbf{x}_{t+1})||q_\phi(\mathbf{y}_t|\mathbf{x}_{t+1})) \quad (22)$$

The initializer $q_\phi(\mathbf{y}_t|\mathbf{x}_{t+1})$, which is a conditional top-down generator, learns to be the stationary distribution of the MCMC transition $K_\theta(\mathbf{x}_t|\mathbf{y}_{t+1})$ by adjusting its mapping towards the low-energy regions of $p_\theta(\mathbf{x}_t|\mathbf{y}_{t+1})$. In a limit, the initializer $q_\phi(\mathbf{y}_t|\mathbf{x}_{t+1})$ minimizes $\text{KL}(K_\theta q_{\phi_j}(\mathbf{y}_t|\mathbf{x}_{t+1})||q_\phi(\mathbf{y}_t|\mathbf{x}_{t+1}))$ and approach the conditional EBM $p_\theta(\mathbf{x}_t|\mathbf{y}_{t+1})$. The entire learning algorithm can be viewed as a chasing game, where the initialzier model $q_\phi(\mathbf{y}_t|\mathbf{x}_{t+1})$ chases the EBM $p_\theta(\mathbf{y}_t|\mathbf{x}_{t+1})$ in pursuit of the true conditional distribution $\pi(\mathbf{y}_t|\mathbf{x}_{t+1})$.

According to the above learning behavior, we can now discuss the benefits of the cooperative learning compared to directly training the initializer with observed $\mathbf{y}$.

Firstly. as presented in Equation 22, the MCMC process of the EBM drives the evolution of the initializer, which seeks to amortize the MCMC. At each learning iteration, in order to provide good initial examples for the current EBM's MCMC, the initialzer needs to be sufficiently close to the EBM. Therefore, training the initialzer with the MCMC outputs $\tilde{\mathbf{y}}$ is a beneficial strategy to maintain a appropriate distance between EBM and initialzer model. Conversely, if the initializer directly learns from the true distribution, even though it may move toward the true distribution quickly, it might not provide a good starting point for the MCMC. A competent initializer should assist in identifying the modes of the EBM. Consider a scenario in which the initizlier model initially shifts toward the true distribution by learning directly from $\mathbf{y}$, but the EBM remains distant. Due to a large divergence between the EBM and the initialzier, the latter may not effectively assist the EBM in generating fair samples, especially with finite-step Lanegevin dynamics. A distant initializer could lead to unstable training of the EBM.

Secondly, let us consider a more general senario where our initializer is modeled using a non-Gaussian generator $\mathbf{y}_t = g_\phi(\mathbf{x}_{t+1}, \mathbf{z}, t)$, where $\mathbf{z} \sim \mathcal{N}(0, \boldsymbol{I})$ introduces randomness through the latent vector $\mathbf{z}$. In this case $q_\phi(\mathbf{y}_t|\mathbf{x}_{t+1}) = \int p(\mathbf{y}_t|\mathbf{x}_{t+1}, \mathbf{z})p(\mathbf{z})d\mathbf{z}$ is analytically intractable. Learning $q_\phi(\mathbf{y}_t|\mathbf{x}_{t+1})$ directly from $\mathbf{y}$ independently requires MCMC inference for the posterior distribution $q_\phi(\mathbf{z}|\mathbf{x}_{t+1}, \mathbf{y}_t)$. However, cooperative learning circumvents the challenge of inferring the latent variables $\mathbf{z}$. That is, at each learning iteration, we generate examples $\hat{\mathbf{y}}$ from $q_\phi(\mathbf{y}_t|\mathbf{x}_{t+1})$ by first sampling $\hat{\mathbf{z}} \sim p(\mathbf{z})$ and then mapping it to $\hat{\mathbf{y}}_t = g_\phi(\mathbf{x}_{t+1}, \hat{\mathbf{z}}, t)$. The $\hat{\mathbf{y}}$ is used to initialze the EBM's MCMC that produces $\tilde{\mathbf{y}}$. The learning equation of $\phi$ is $\frac{1}{n}\sum_{i=1}^{n} -\frac{1}{2\bar{\sigma}_t{}^2}||\tilde{\mathbf{y}}_{t,i} - g_\phi(\mathbf{x}_{t+1,i}, \hat{\mathbf{z}}, t)||^2$, where the latent variables $\hat{\mathbf{z}}$ is used. That is, we shift the mapping from $\hat{\mathbf{z}} \rightarrow \hat{\mathbf{y}}$ to $\hat{\mathbf{z}} \rightarrow \tilde{\mathbf{y}}$ for accumulating the MCMC transition. Although our paper currently employs a Gaussian initializer, if we adopt a more expressive non-Gaussian initializer in the future, the current cooperative learning strategy (i.e., training $q_\phi$ with $\tilde{\mathbf{y}}$) can be much more beneficial and feasible.

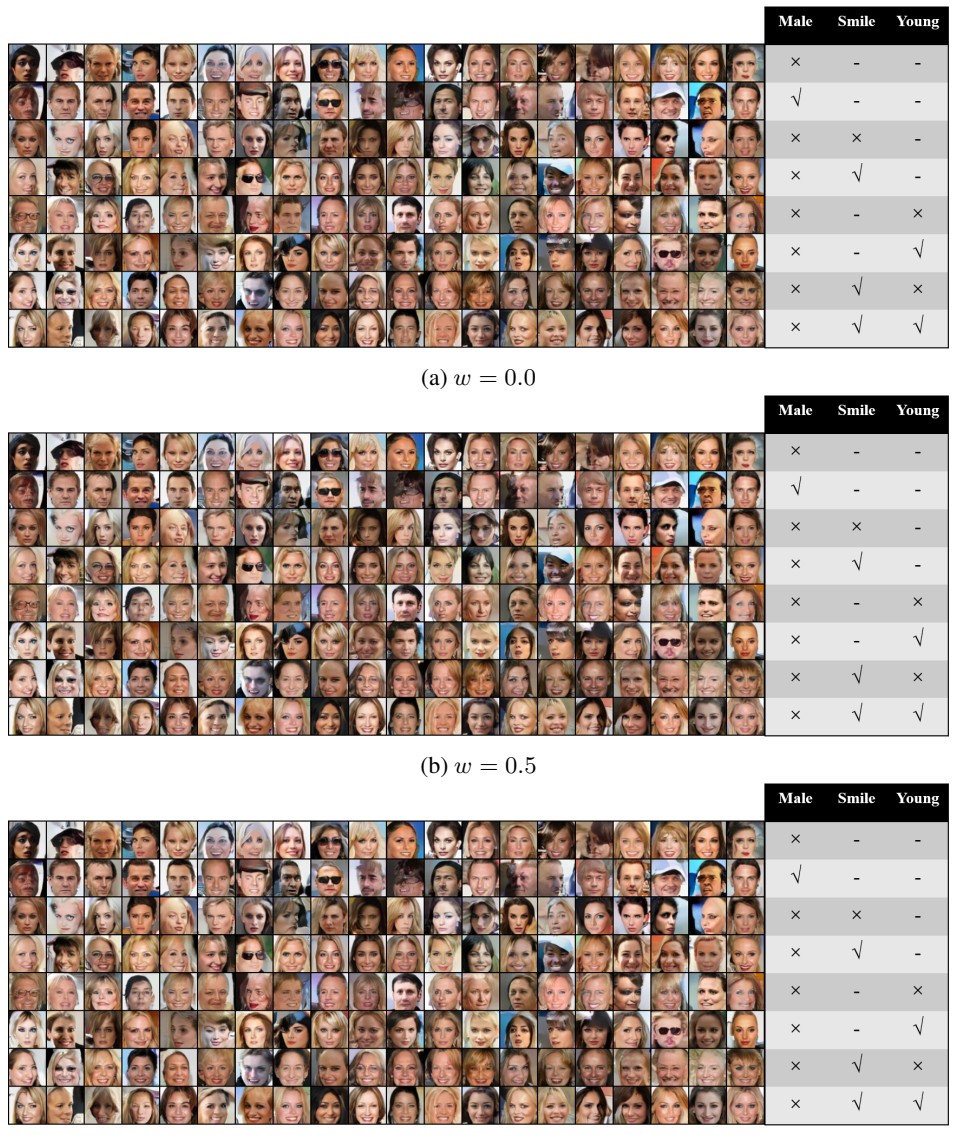

Figure 10: Attribute compositional samples generated by CDRL models trained on the CelebA ($64 \times 64$) dataset. We utilize guided weights $w = 0.0, 0.5, 1.0$. Images at different guidance share the same random noise. Results can also be compared with those in Figure 4, which use $w = 3.0$.

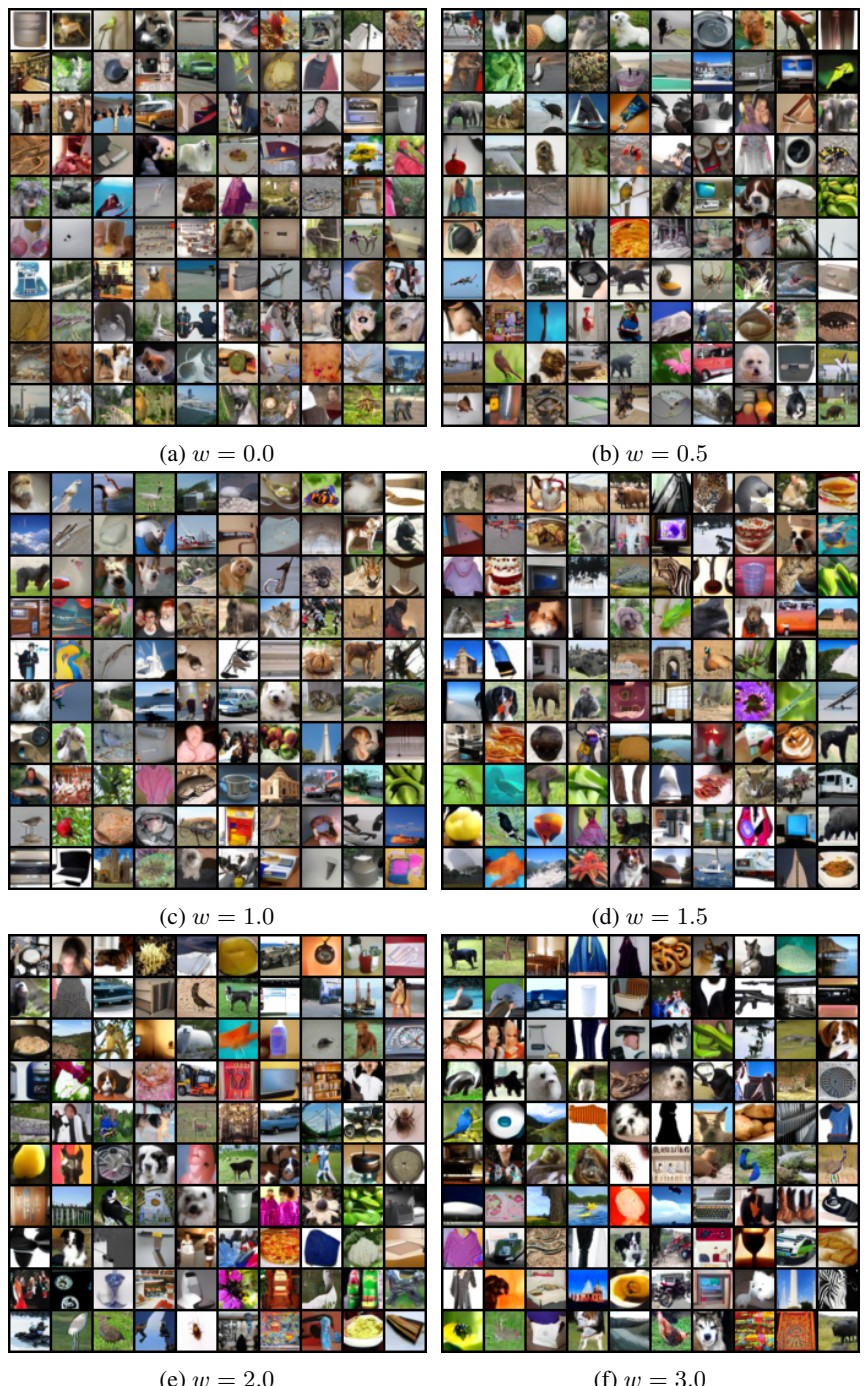

(a) $w = 0.0$        (b) $w = 0.5$

(c) $w = 1.0$        (d) $w = 1.5$

(e) $w = 2.0$        (f) $w = 3.0$

Figure 11: Conditional generated examples with various classifier-free guidance weights on the ImageNet32 ($32 \times 32$) dataset. Samples are generated using randomly selected class labels.

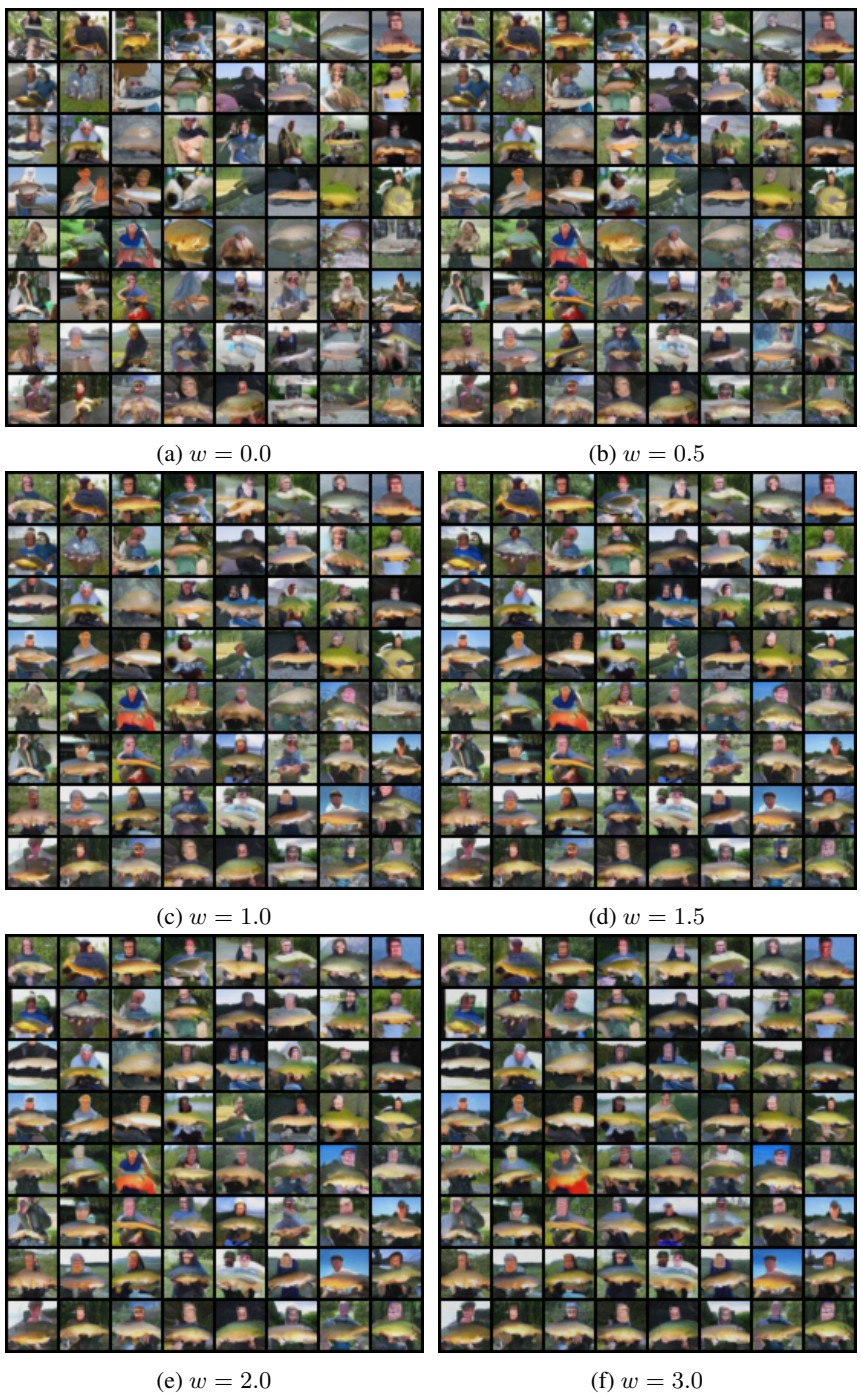

(a) $w = 0.0$            (b) $w = 0.5$

(c) $w = 1.0$            (d) $w = 1.5$

(e) $w = 2.0$            (f) $w = 3.0$

Figure 12: Conditional generated examples with different classifier-free guidance weights on the ImageNet32 ($32 \times 32$) dataset, using the class label "Tench".

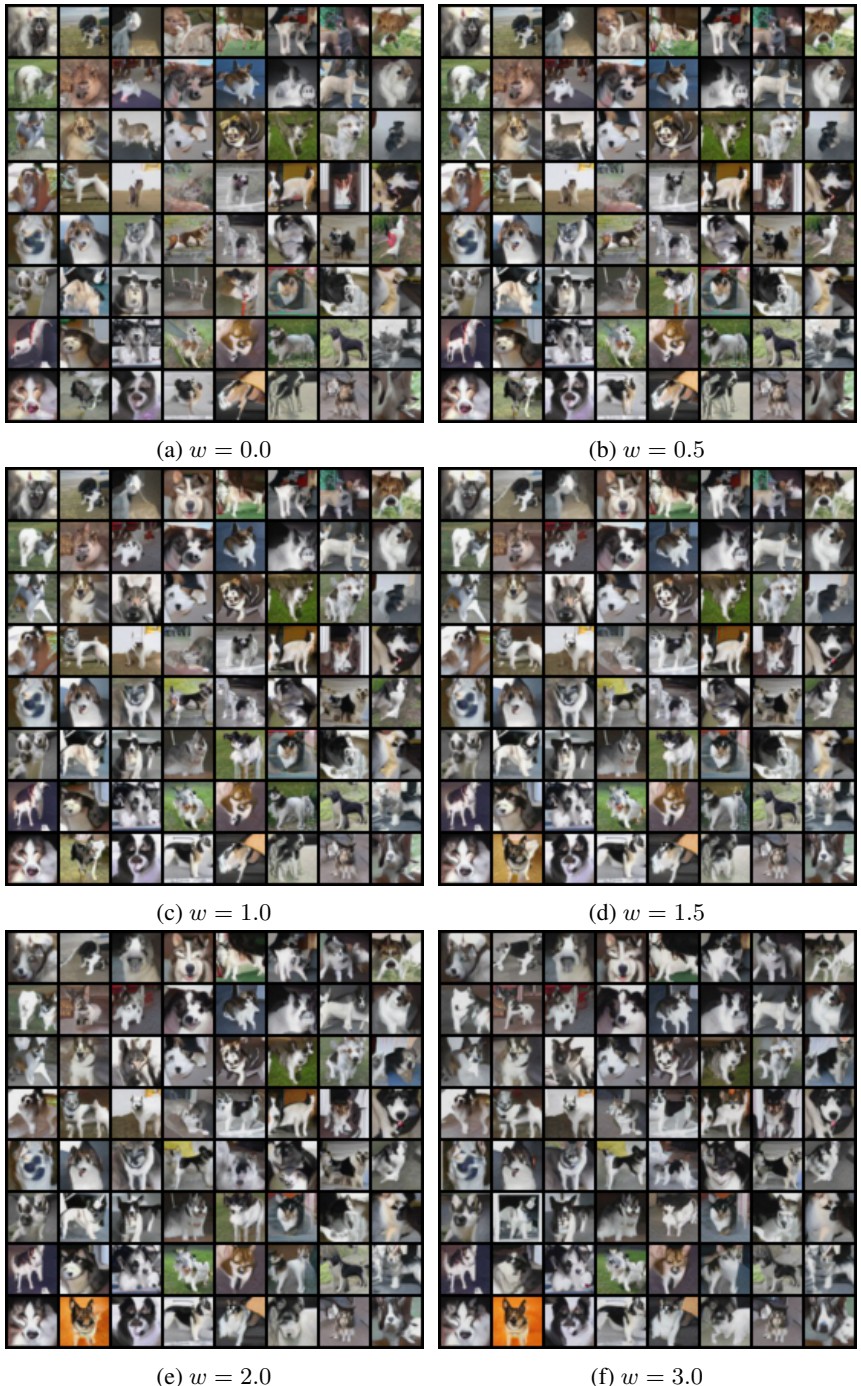

(a) $w = 0.0$      (b) $w = 0.5$

(c) $w = 1.0$      (d) $w = 1.5$

(e) $w = 2.0$      (f) $w = 3.0$

Figure 13: Conditional generated examples with different classifier-free guidance weights on the ImageNet32 ($32 \times 32$) dataset, using the class label "Siberian Husky".

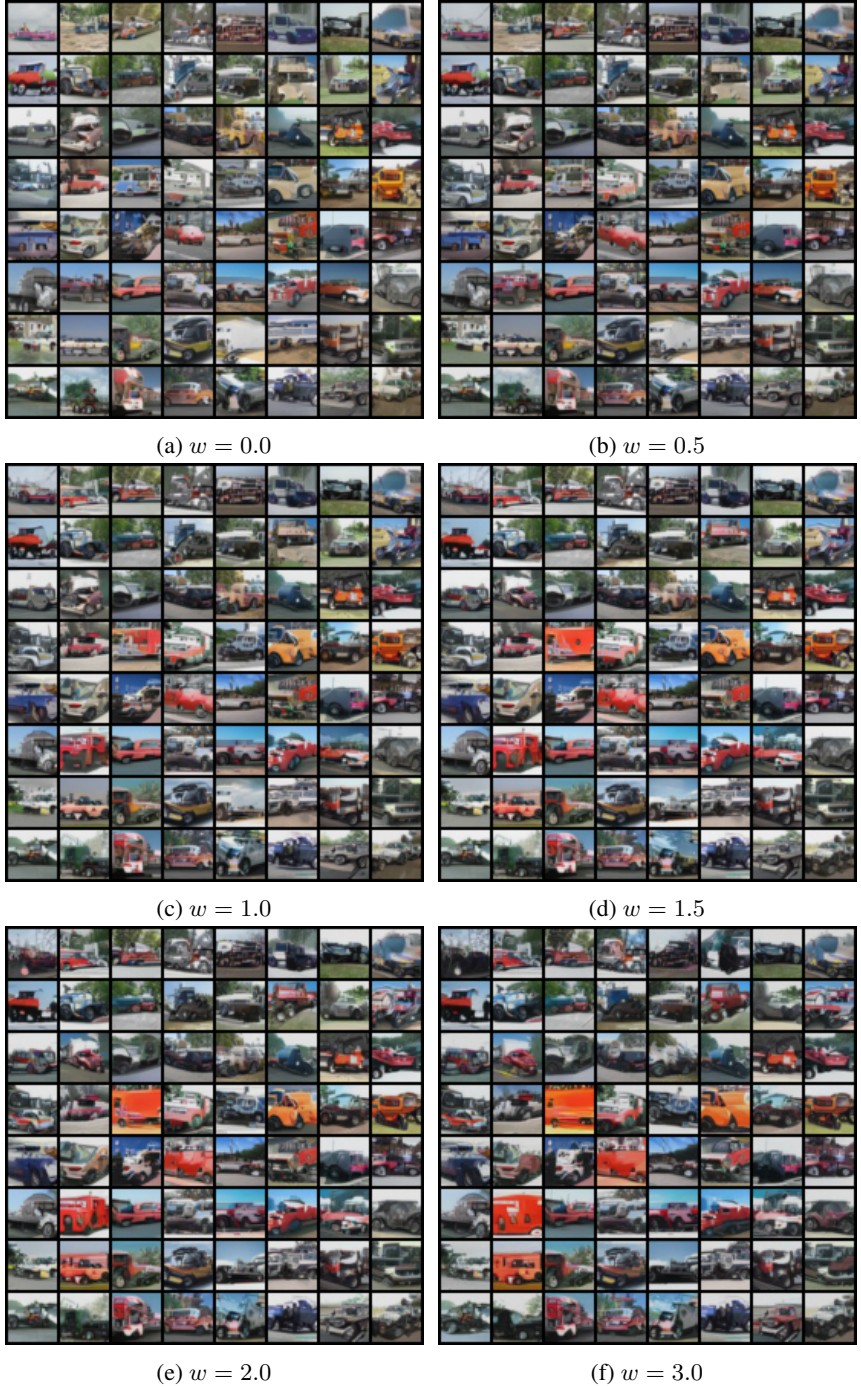

(a) $w = 0.0$

(b) $w = 0.5$

(c) $w = 1.0$

(d) $w = 1.5$

(e) $w = 2.0$

(f) $w = 3.0$

Figure 14: Conditional generated examples with different classifier-free guidance weights on the ImageNet32 ($32 \times 32$) dataset, using the class label "Tow Truck".

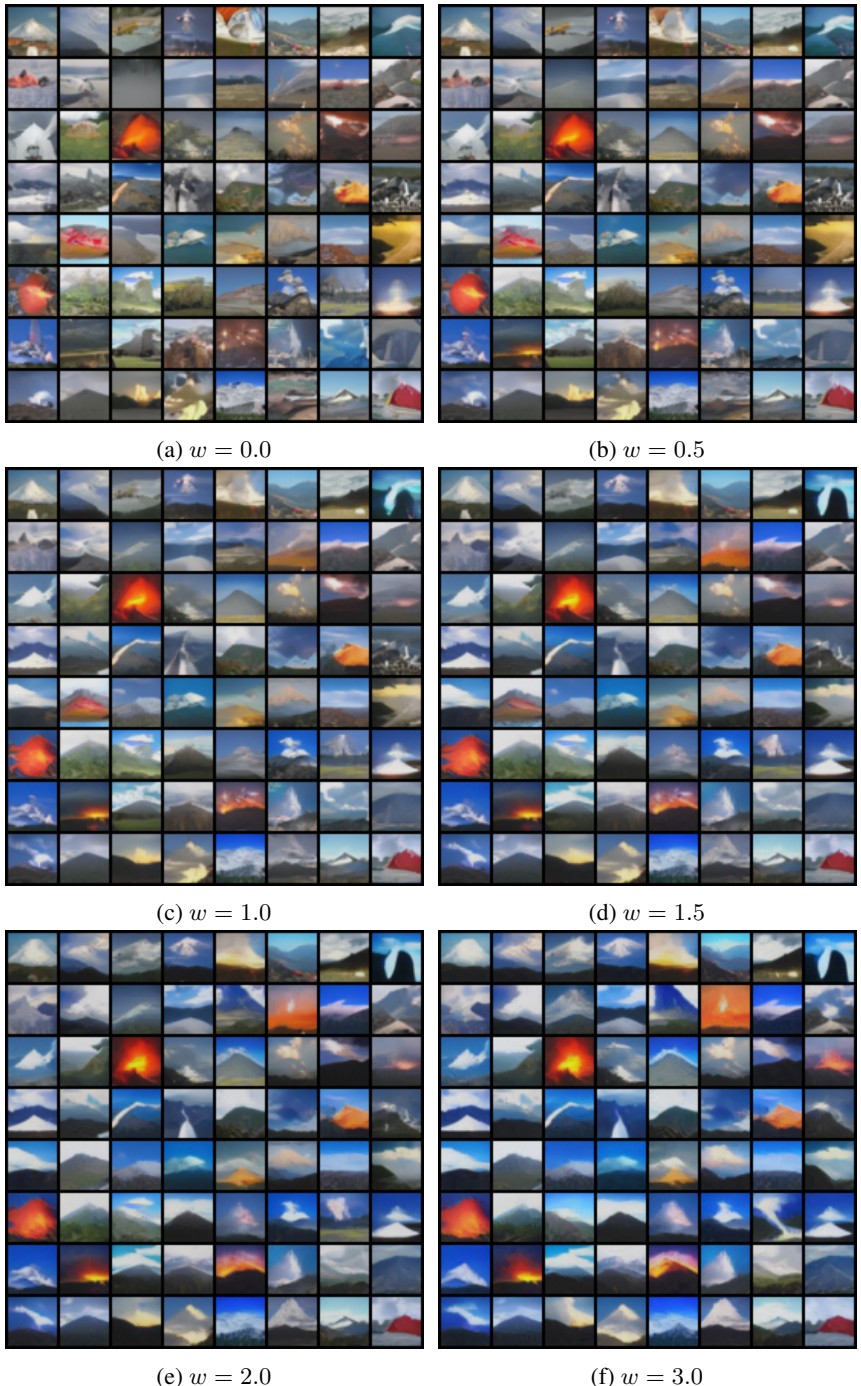

(a) $w = 0.0$

(b) $w = 0.5$

(c) $w = 1.0$

(d) $w = 1.5$

(e) $w = 2.0$

(f) $w = 3.0$

Figure 15: Conditional generated examples with different classifier-free guidance weights on the ImageNet32 ($32 \times 32$) dataset, using the class label "Volcano".

