# OpenReview forum: "Learning Energy-Based Models by Cooperative Diffusion Recovery Likelihood"
_ICLR.cc/2024/Conference — ICLR 2024 spotlight_

### Official Review · Reviewer_KQ36 · 2023-10-27

**Soundness:** 3 good
**Presentation:** 2 fair
**Contribution:** 3 good
**Rating:** 8
**Confidence:** 3

**Summary:**

The authors study methods for training and sampling from energy-based models (EBMs).

They propose Cooperative Diffusion Recovery Likelihood (CDRL), which is an extension of Diffusion Recovery Likelihood (DRL). The proposed CDLR aims to improve the sample quality of DLR, while also reducing the number of MCMC steps required during training and sampling. CDLR entails jointly estimating a sequence of EBMs and MCMC initializers, utilizing the cooperative training approach. Essentially, the main proposed method is a combination of DLR (Gao et al., 2021) and cooperative training (Xie et al., 2020).

The proposed method is shown to improve the sample quality of DLR in unconditional generation on CIFAR10 and ImageNet 32x32. It is also applied to some conditional generation (via classifier-free guidance), compositional generation and OOD detection experiments.

**Strengths:**

The paper is well written overall. I found it interesting to read. The authors definitely seem knowledgeable and familiar with previous work.

The main idea of the proposed method, combining DRL with cooperative training, makes intuitive sense and is described well overall.

The proposed method seems to improve the DLR baseline in experiments.

**Weaknesses:**

The related work section is placed in the Appendix, which seems quite odd. As a result, the previous work on cooperative training is sort of "hidden" in the main paper. This should be described before Section 3.3.

The similarities and differences between the proposed approach using EBMs and diffusion models could be discussed in more detail. The "Diffusion Model" paragraph in the related work section is interesting, but I think this could be expanded on a lot more.

**Questions:**

1. I think that the proposed CDLR method makes sense, and that it seems like a promising method for improving EBM training/sampling. However, it is not entirely clear to me how the resulting EBM differs from diffusion models? What are the main pros and cons compared to diffusion models? Why should one use this type of EBM instead of a diffusion model? When / in which applications should one use this type of EBM?

2. Would it be possible to compare CDLR with the DLR baseline also in the OOD detection experiment (Table 4)?

3. Would it perhaps be possible to illustrate / give a schematic overview of the proposed approach in some kind of figure? (a sequence of noise levels, one EBM and initializer model per level etc. I just think that this perhaps could help illustrate the general idea)

4. The experiment in Figure 7 is interesting. Could you report FID scores for (a) and (b)? I.e., for K=0 and K=15 steps of Langevin refinement. Could this experiment perhaps also be repeated for K=1, 2, 4, 8 steps of Langevin refinement?



Minor things:
- Section 3.1, "...constrains the conditional energy landscape to be localized around y_t": Should y_t be x_{t+1}?
- Abstract, "noisy versons": versons --> versions.
- Section 1, "The initializer model proposes initial samples by making prediction of the samples at the current noise level given": "making a prediction"? "making predictions"? "by predicting the samples..."?
- Section 3.2, "which may again requires MCMC sampling": requires --> require?
- Algorithm 1: noise level --> noise levels, sampling step --> sampling steps.
- Algorithm 2: sampling steps L --> sampling steps K? (also some inconsistency with this in the Appendix)
- 4.4: CDLR strong --> CDLR achieves strong?
- G.5: doesn't gives --> doesn't give (or, "does not give", I suppose)?

---

> ### Author Response · Authors · 2023-11-22
> **Thank you for your detailed comments and constructive suggestions. (1/2)**
>
> Thank you for thoroughly reviewing our paper and providing constructive feedback. We hope that the revisions made in accordance with your suggestions have clarified key points, and that our responses address your concerns effectively
>
> ### 1. The related work section is placed in the Appendix, which seems quite odd. As a result, the previous work on cooperative training is sort of "hidden" in the main paper. This should be described before Section 3.3.
> Thank you for your advice. Our arrangement was primarily influenced by the page limit constraints. We believed it would be more beneficial to allocate more space to experimental results in the main body of the paper, while relegating a detailed discussion of related work to the appendix. Following your suggestions, we have expanded our discussion in Section 3.2 to further explore the connection between our work and cooperative training. Additionally, we guide readers to the appendix for more comprehensive discussions.
>
> &nbsp;
>
> ### 2. The similarities and differences between the proposed approach using EBMs and diffusion models could be discussed in more detail. The "Diffusion Model" paragraph in the related work section is interesting, but I think this could be expanded on a lot more.
>
> Thank you for the suggestion, we have added more discussion on this topic to the modified version of our paper. (Please see page 20) Here is our current discussion for diffusion model:
>
> "Diffusion models, originating from Sohl-Dickstein et al. (2015) and further developed in works such as Song & Ermon (2020); Ho et al. (2020), generate samples by progressively denoising them from a high noise level to clean data. These models have achieved remarkable success in generating high-quality samples from complex distributions, thanks to various architectural and framework innovations Ho et al. (2020); Song et al.; Kim et al. (2021); Song et al. (2021); Dhariwal & Nichol (2021); Karras et al. (2022); Ho & Salimans (2022). Notably, Dhariwal & Nichol (2021) emphasizes that the generative performance of diffusion models can be enhanced with the aid of a classifier, while Ho & Salimans (2022) further demonstrates that this guided scoring can be estimated by the differential scores of a conditional model versus an unconditional model. Enhancements in sampling speed have been realized through distillation techniques Salimans & Ho (2022) and the development of fast SDE/ODE samplers Song et al.; Karras et al. (2022); Lu et al. (2022a). Recent advancements Rombach et al. (2022); Saharia et al. (2022); Ramesh et al. (2022) have successfully applied conditional diffusion models to the task of text-to-image generation, achieving significant breakthroughs.
>
> EBM shares a close relationship with diffusion models, as both frameworks can provide a score to guide the generation process, whether through Langevin dynamics or SDE/ODE solvers. As Salimans & Ho (2021) discusses, the distinction between the two lies in their implementation approach: EBMs model the log-likelihood directly, while diffusion models concentrate on the gradient of the log-likelihood. This distinction enables EBM to be used in some potential applications. These include utilizing advanced sampling techniques Du et al. (2023), transformed into classifiers Guo et al. (2023), or employed in the detection of abnormal samples through estimated likelihood Grathwohl et al. (2020); Liu et al. (2020).
>
> The focus of this work is to push the development of EBM. And our work connects to diffusion models (Ho et al., 2020; Xiao et al., 2022) by learning a sequence of EBMs and MCMC initializers to reverse the diffusion process. Contrasting (Ho et al., 2020), our framework employs more expressive conditional EBMs instead of normal distributions. (Xiao et al., 2022) also suggests multimodal distributions, trained by generative adversarial networks (Goodfellow et al., 2020), for the reverse process. "

---

> ### Author Response · Authors · 2023-11-22
> **Thank you for your detailed comments and constructive suggestions. (2/2)**
>
> ### 3. I think that the proposed CDLR method makes sense, and that it seems like a promising method for improving EBM training/sampling. However, it is not entirely clear to me how the resulting EBM differs from diffusion models? What are the main pros and cons compared to diffusion models? Why should one use this type of EBM instead of a diffusion model? When / in which applications should one use this type of EBM?
>
> Thank you for your question. There is a close connection between EBM and diffusion models. Both frameworks offer scores to guide the generation process, whether through Langevin dynamics or SDE/ODE solvers. The key distinction lies in their implementation: EBM models the log-likelihood, whereas diffusion models focus on the gradient of the log-likelihood. We appreciate the significant success achieved by diffusion models. But we believe EBMs also demonstrate substantial applicability, as evidenced by numerous use cases. For example, in composition generation tasks, one study [2] highlights the benefits of using an energy function with advanced samplers like HMC to achieve improved results. In Simulation-Based Inference, another research [3] employs a fitted function for neural likelihood estimation, and continuous learning applications [4] leverage the energy function for model fusion. Furthermore, EBMs are used to model set functions [5], showcasing their versatility across various domains. And we have discussed more applications (e.g OOD detection, transform into classifier) in our introduction or related works.
>
> We believe in the untapped potential of EBMs, either as a powerful generative models or in applications requiring complex distribution likelihood estimation. However, modeling intricate distributions like images remains a challenge for EBMs, limiting their performance in generation and likelihood estimation. This limitation largely stems from the absence of efficient sampling strategies for stable training. Our research advances EBM studies by devising a simple yet effective training methodology. We implement practical design decisions concerning the sampling process, network architecture, and noise variance reduction. While the techniques provided in one single work may not be the ultimate solution to all the challenges, we are confident it significantly contributes to EBM research.
>
>
> [1] Learning energy based models by diffusion recovery likelihood.
>
> [2] Reduce, Reuse, Recycle: Compositional Generation with Energy -Based Diffusion Models and MCMC.
>
> [3] Maximum Likelihood Learning of Energy-Based Models for Simulation-Based Inference.
>
> [4] Beef: Bi-compatible class-incremental learning via energy-based expansion and fusion.
>
> [5] Learning Neural Set Functions Under the Optimal Subset Oracle.
>
>
>
> ### 4. Would it be possible to compare CDLR with the DLR baseline also in the OOD detection experiment (Table 4)?
>
> Sorry, given the limited time of the discussion period, we can not retrain a model for DRL and then apply it to OOD. Instead, after searching among the recent works, we find that [1] has reported the the score for DRL. According to their observations, DRL achieves auroc score of 0.4377 on Cifar100 (vs 0.78 for CDRL) as OOD dataset and 0.6398 on CelebA dataset (vs 0.84 for CDRL). According to this, we can see that CDRL achieves better OOD detection score than DRL.
>
> [1] Energy-Based Models for Anomaly Detection: A Manifold Diffusion Recovery Approach
>
> &nbsp;
>
> ### 5. Would it perhaps be possible to illustrate / give a schematic overview of the proposed approach in some kind of figure? (a sequence of noise levels, one EBM and initializer model per level etc. I just think that this perhaps could help illustrate the general idea)
>
> Thank you for your advice. We have added an overall figure to the revised version of our paper. Please see Figure 5 (a) and (b) in the modified version of our paper.
>
> &nbsp;
>
> ### 6. The experiment in Figure 7 is interesting. Could you report FID scores for (a) and (b)? I.e., for K=0 and K=15 steps of Langevin refinement. Could this experiment perhaps also be repeated for K=1, 2, 4, 8 steps of Langevin refinement?
>
> Certainly. In accordance with your advice, we tested the FID for Figure 7 (current Figure 8) using various Langevin steps. The outcomes are presented in the table below. It's important to note that for this particular setting, we merely reduced the number of Langevin steps without adjusting their step size as detailed in Section 4.2.
>
> Table: FID scores for ImageNet32 unconditional generation with different number of Langevin steps
>
> |  Total MCMC steps |  FID |
> | ----------------- |------|
> |K=0| 38.94|
> |K=1| 34.97|
> |K=2| 30.89|
> |K=4| 24.75|
> |K=8| 16.21|
> |K=15| 9.35|
>
> &nbsp;
>
> ### 7. Minor typo:
>
> Thank you very much for helping us find the typos, we have modified in the current version of our paper.

---

> > ### Comment · Reviewer_KQ36 · 2023-11-22
> > **Response to rebuttal**
> >
> > (Sorry for my late response. I have struggled to find enough time to both write responses as an author, and participate in the discussions as a reviewer)
> >
> > I have read the other reviews and all responses.
> >
> > The authors provided a detailed and thorough rebuttal. Most of my questions have been addressed well. The added overview figure in Figure 5 is helpful, thank you. The results in answer 6. above are neat.
> >
> > The answer to my Question 1 could have been more specific, I agree with Reviewer cEiK that "the paper could have been better if the authors had compared CDRL with diffusion in the mentioned tasks", but I still think this is a good paper overall.
> >
> > I have increased my score from "6: marginally above" to "8: accept".

---

> ### Author Response · Authors · 2023-11-23
> **Thank your for your comments and suggestions!**
>
> Dear Reviewer  KQ36:
>
> We sincerely appreciate your insightful comments and are grateful for the time you dedicated to reviewing our paper and providing constructive feedback, particularly during this challenging rebuttal and discussion period. Your advice on related work and the overall scheme has made our paper more clear.
>
> As mentioned in our previous response, we are enthusiastic about the untapped potential of EBMs. Therefore, we wholeheartedly agree and are eager to implement EBM (and our CDRL training) in more real-world applications. Given the limited discussion period and considering that the primary focus of this paper is to propose a general training method, we plan to explore these exciting directions in our future work.

---

### Official Review · Reviewer_S2Sh · 2023-10-29

**Soundness:** 3 good
**Presentation:** 2 fair
**Contribution:** 2 fair
**Rating:** 6
**Confidence:** 5

**Summary:**

The authors propose a new method for learning energy-based models (EBMs) mimicking diffusion models. Specifically, within the diffusion-model framework, EBMs are modified to parameterize the denoising process; to accelerate the MCMC sampling process when training the EBM, the authors also use a simultaneously trained diffusion-like model as an ``initialzer model.'' Experimental results demonstrate the effectiveness of the proposed method.

**Strengths:**

The presented techniques are likely new.

**Weaknesses:**

The clarity should be improved. For example, consider the relationships between the proposed method and related works.

The contribution is kind of incremental. Based on the Diffusion Recovery Likelihood (DRL) (Gao et al., 2021) in Sec. 3.1, the authors proposed in Sec. 3.2 a new "initializer model" to amortize the expensive MCMC sampling.

**Questions:**

1. What are the main contributions of the proposed CDRL when compared with DRL (Gao et al., 2021)?

2. In the paragraph following Eq. (5), why "maximizing recovery likelihood still guarantees an unbiased estimator of the true parameters of the marginal distribution of the data?"

3. Eqs. (6) and (8) are quite like the fomula of the conventional diffusion model. Please elaborate on the relationships between them.

4. It seems that Eq. (9) indicates a deterministic noising process, right? If so, why "there’s still variance for xt given x0 and xt+1?"

5. How to interpret Eq. (11), especially the $\tilde p_{\theta}$?

---

> ### Author Response · Authors · 2023-11-22
> **Thank you for your comments and questions. (1/2)**
>
> Thank your for your comments and questions. Here are our responses. Hope they makes things more clear and can solve your concerns.
>
> ### 1. The relationship between our work and previous works. The contribution is kind of incremental.
>
> Thank you for your question. The goal of our work is to push the study of Energy Based Model. We would like to emphasize that, while the study of EBM draws more and more attentions recently, unlike the popular diffusion model, the scaling up of training for EBM is still a largely uncharted territory. The primary challenge lies in devising a more efficient sampling strategy. In pursuit of this goal, many individual efforts have been made. For instance, DRL[1] employs a series of noise levels, and cooperative learning[2, 3] introduces a generator or initializer. Though these methods have led to improvements in their own contexts, they are often regarded as separate techniques, and their individual performance may exhibit noticeable gaps when compared to other generative frameworks. We are the first in integrating these elements into a unified framework, and we are the first to successfully demonstrate that by combining these two techniques, the performance of EBM can be substantially enhanced.
>
> Furthermore, we make many practical design choices related to noise scheduling, network architecture, noise variance reduction. These choices helps the training of EBM and we carefully conducted ablation studies on each component (please see our Appendix G). We believe these designs can greatly benefit future works in the study of EBM.
>
> Besides, we show that EBM trained with CDRL can benefit from simple sampling adjustment techniques to greatly reduce its sampling steps. This further improves the sampling efficiency of EBM as a generative model and might inspire future works in similar directions.
>
> Thus, we believe that our work has made great contributions. Our model and algorithm can pave the way for the scaling up of EBM training on complex distributions.
>
> [1] Learning Energy-Based Models by Diffusion Recovery Likelihood
>
> [2] Cooperative training of descriptor and generator networks.
>
> [3] A tale of two flows: Cooperative learning of langevin flow and normalizing flow toward energy-based model.
>
> &nbsp;
>
> ### 2. In the paragraph following Eq. (5), why "maximizing recovery likelihood still guarantees an unbiased estimator of the true parameters of the marginal distribution of the data?"
> The recovery likelihood formulation uses EBM to model the marginal distribution of the data and uses classic MLE loss to optimize the parameter. According to the classical analysis of MLE, the point estimate given by optimizing recovery likelihood model is an unbiased estimator of the true parameter. For more details, please check the appendix A.2 in [1].
>
> [1]Learning Energy-Based Models by Diffusion Recovery Likelihood
>
> &nbsp;
>
> ### 3. Eqs. (6) and (8) are quite like the fomula of the conventional diffusion model. Please elaborate on the relationships between them.
> Inspired by the diffusion model, DRL [1] introduces a method to learn a series of Energy-Based Models (EBMs) across various noise levels denoted as t. Our research expands on DRL by incorporating an initializer at each noise level, enhancing the EBM learning process. Like the diffusion model, our initializer is based on Gaussian distributions. A notable difference, however, is that unlike diffusion models that let the Unet to learn from the data, in our cooperative algorithm, the U-Net is trained using the revised sample of the EBM, i.e. $\tilde{y_t}$ not merely the original data augmented with Gaussian noise; it is the product of a Langevin sampling process, guided by the energy function. This enables the unet to armortized the Langevin sampling process. We delve deeper into this difference and present an ablation study in Appendix G.2, offering further insights into this aspect.

---

> ### Author Response · Authors · 2023-11-22
> **Thank you for your comments and questions. (2/2)**
>
> ### 4. It seems that Eq. (9) indicates a deterministic noising process, right? If so, why "there’s still variance for xt given x0 and xt+1?"
>
> With the noise variance reduction technique, at each iteration, we use the same noise to inject $x_t$ and $x_{t+1}$. However, our model contains both an EBM and its initializer. During the sampling process, at each noise level, the initializer proposes an initialization and the EBM modifies it with MCMC. In both the initial proposal stage and MCMC stage, random noise is injected. During the training process, as discussed in the previous question, our initializer learns from the revised sample of the MCMC process instead of directly learning from the data. This implies that $\tilde{y_t}$ is not completely deterministic given $x_0$ and $x_{t+1}$. On the other hand, we define the EBM on the marginal distribution of $x_t$ (more precisely $y_t$), so updating the EBM only requires drawing samples from the marginal distribution $p(x_t)$, not the joint distribution $p(x_t, x_{t+1})$. The noise variance reduction technique does not alter the marginal distributions of $x_t$ and $x_{t+1}$, thereby ensuring it does not introduce extra bias into EBM training.
>
> &nbsp;
>
> ### 5. How to interpret Eq. (11), especially the $\tilde{p_\theta}$.
>
> Equation (11) is the equation of classifier-free guidance [2]. The classifier-free guidance technique aims to improve the quality of samples in conditional generation, building upon the concept of classifier guidance [1]. In this method, an additional classifier $p(y|x)$ is used in conjunction with eiter a conditional model $p(x|y)$ or an unconditional model $p(x)$  to enhance the estimation of conditional generation scores. For detailed insights, please refer to section 4 in [1]. Further, [2] proposes that instead of employing an additional classifier $p(y|x)$, it might be more effective to estimate the guided score by analyzing the difference between the conditional model $p(x|y)$ and the unconditional model $p(x)$. Equation (11) is derived following the methodologies in [1, 2]. Under our notation system, in Equation 11, $f_\theta (y_t; c, t) = \log p_\theta(y_t|c)$ denotes the score for conditional generation, and $f_\theta (y_t; t) = \log p_\theta(y_t)$ represents the score for the unconditional model. Instead of directly using $\log p_\theta(y_t|c)$ in the sampling process, we utilize an enhanced version $\tilde{p_\theta}(y_t|c)$, which combines aspects of both the conditional and unconditional models, in a manner similar to [2].
>
> [1] Diffusion Models Beat GANs on Image Synthesis
>
> [2] Classifier-Free Diffusion Guidance

---

> > ### Author Response · Authors · 2023-11-23
> >
> > Dear Reviewer S2Sh,
> >
> > We are sincerely grateful for the time and effort you have invested in reviewing our paper and for your valuable comments during this constrained timeframe. We would like to kindly ask if you have any additional concerns regarding our work. Should you have any further inquiries or require additional clarification, we are happy to provide further clarifications.

---

> > > ### Comment · Reviewer_S2Sh · 2023-11-23
> > >
> > > Thanks for the detailed responses. I have updated my ratings.

---

> > > > ### Author Response · Authors · 2023-11-23
> > > >
> > > > Dear reviewer S2Sh:
> > > >
> > > > Thank you once more for your insightful comments and questions, as well as for responding to our feedback and adjusting your score during this short discussion period. Your support and reviews are sincerely appreciated.

---

### Official Review · Reviewer_nfgg · 2023-11-03

**Soundness:** 3 good
**Presentation:** 3 good
**Contribution:** 2 fair
**Rating:** 6
**Confidence:** 4

**Summary:**

This paper proposes the Cooperative Diffusion Recovery Likelihood (CDRL) for training energy-based models. Compared with the baseline DRL, CDRL introduces an extra initializer model that is jointly trained with the EBM along the diffusion process. The required MCMC steps are reduced thanks to a better initial sample.
The experiments show that CDRL has significantly improved over the baseline DRL method.

Besides the proposed method, the paper also discusses some other aspects of EBMs such as noise scheduler designs, classifier-free guidance, etc.

**Strengths:**

The paper is well-organized and easy to follow.
I think the highlight of this paper is the strong empirical performance of CDRL. The work shows the possibility of achieving an FID on image benchmarks at least comparable to other generative models such as GANs and diffusion models.
The ablation studies are convincing as validation of each of the components in the proposed method.

Besides, the introduced classifier-free guidance for EBM is also empirically sound, which verifies many intuitions from diffusion models.

**Weaknesses:**

My main concern is the lack of novelty.
The key components are inspired by other works. The DRL is well-established in training EBMs and the idea of a trainable initializer for MCMC is also not new. The combination of them seems a bit hacky to me and the overall method is a little cumbersome.
It would be better if the motivation and necessity of introducing an initializer (because initializers bring additional costs) were explained more clearly with more new insights to DRL.

Nonetheless, making this idea work empirically is impressive and the generative performance of EBMs has been greatly improved.
However, the image generation experiment is on very low resolutions. I think it would be more convincing if the method could be shown to be able to handle image generation tasks for higher resolutions;

**Questions:**

The performance gain, regardless of efficiency, seems to come from a better initial state provided by the cooperatively trained initializer. In this case, I wonder about the limit of the baseline CRL, if we increase the MCMC sampling steps significantly, say 300, what would the performance be?

**Details Of Ethics Concerns:**

No ethics concerns

---

> ### Author Response · Authors · 2023-11-22
> **Thank you for your comments! (1/2)**
>
> Thank you for your review. We appreciate your comments "making this idea work empirically is impressive and the generative performance of EBMs has been greatly improved." Please see our responses below, hope they can solve your concerns.
>
> ### 1.The key components are inspired by other works.
>
> Our research aims to advance the study of EBM. We observe that while EBMs are gathering increasing interest, scaling up their training, unlike with popular diffusion models, remains largely unexplored. The main challenge is developing a more efficient sampling strategy. Various efforts have been made in this direction. For example, the DRL approach incorporates multiple noise levels in EBM, achieving significant improvements and establishing a robust baseline. Nonetheless, as mentioned in the DRL study, further enhancing performance through using more sampling steps or noise levels proves challenging, with a noticeable performance disparity between DRL and other generative models. Designing superior sampling algorithms continues to be a complex endeavor. Although several options, such as amortizing MCMC, exist in the literature, selecting the appropriate technique and developing an improved algorithm is not straightforward. Our work is pioneering in assimilating these elements into a cohesive framework. We are the first to demonstrate that by merging these techniques, the performance of EBMs can be markedly improved.
>
> Furthermore, we make many practical design choices related to noise scheduling, network architecture, noise variance reduction. These choices helps the training of EBM and we carefully conducted ablation studies on each component (please see our Appendix G). We believe these designs can greatly benefit future works in the study of EBM.
>
> Besides, we show that EBM trained with CDRL can benefit from simple sampling adjustment techniques to greatly reduce its sampling steps. This further improves the sampling efficiency of EBM as a generative model and might inspire future works in similar directions.
>
> Thus, we believe that our work has made great contributions. Our model and algorithm can pave the way for the scaling up of EBM training on complex distributions.
>
> &nbsp;
>
> ### 2. Nonetheless, making this idea work empirically is impressive and the generative performance of EBMs has been greatly improved. However, the image generation experiment is on very low resolutions. I think it would be more convincing if the method could be shown to be able to handle image generation tasks for higher resolutions;
>
> Thank you for your comments. We have observed that recent trends in generative modeling for high-resolution images involve either leveraging the latent space of a VAE, as exemplified by Latent Diffusion[5], or by initially creating a low-resolution image which is then upscaled, a method employed by models like Imagen[6]. These approaches often compresses the modeled space into dimensions such as 32 × 32 or 64 × 64, which is consistent with the resolution used in our experiments. Following your suggestions, we conducted an additional experiment during the rebuttal phases where we trained an Energy-Based Model within the latent space of a latent diffusion model using the CelebA-HQ dataset, which has a resolution of 256 x 256. Our preliminary results have achieved an FID score of 10.74. We have included the new results in our modifed paper (see Figure 9 in the Appendix). In the following Table, we compare our preliminary results with some baselines on this dataset.
>
> Table FID score for CelebA-HQ (256 x 256) dataset
> |Model | FID score|
> |------| ---------|
> |GLOW[3]| 68.93|
> |VAEBM[1] |20.38|
> |ATEBM [2]|17.31 |
> |VQGAN+Transformer[4]| 10.2|
> |LDM[5]|5.11|
> |CDRL(ours)  |10.74|
>
> [1] VAEBM: A Symbiosis between Variational Autoencoders and Energy-based Models
>
> [2] Learning energy-based models with adversarial training
>
> [3] Glow: Generative Flow with Invertible 1x1 Convolutions
>
> [4] Taming Transformers for High-Resolution Image Synthesis
>
> [5] High-Resolution Image Synthesis with Latent Diffusion Models
>
> [6] Photorealistic Text-to-Image Diffusion Models with Deep Language Understanding

---

> > ### Author Response · Authors · 2023-11-22
> > **Thank you for your comments! (2/2)**
> >
> > ### 3. The performance gain, regardless of efficiency, seems to come from a better initial state provided by the cooperatively trained initializer. In this case, I wonder about the limit of the baseline CRL, if we increase the MCMC sampling steps significantly, say 300, what would the performance be?
> >
> > We believe that by "baseline CRL," you are referring to "baseline DRL" [1]. As detailed in [1] (Table 2), it's important to note that merely increasing the number of MCMC steps in DRL has a limited impact on generative performance. Specifically, increasing the sampling step T from 30 to 50 in DRL only slightly improves the FID score, from 9.58 to 9.36, while significantly increasing the sampling time. We anticipate a similar effect if MCMC sampling steps were to be further increased to 300, though this might be computationally impractical for training. Conversely, by incorporating an initializer, our CDRL notably enhances the sampling efficiency of DRL and simultaneously reduces the sampling time, underscoring the significant contribution of our work.
> >
> > [1] Learning Energy-Based Models by Diffusion Recovery Likelihood

---

> > > ### Author Response · Authors · 2023-11-23
> > >
> > > Dear Reviewer nfgg,
> > >
> > > We are sincerely grateful for the time and effort you have invested in reviewing our paper and for your valuable comments during this constrained timeframe. We would like to kindly ask if you have any additional concerns regarding our work. Should you have any further inquiries or require additional clarification, we are happy to provide further clarifications.

---

> > > > ### Comment · Reviewer_nfgg · 2023-11-23
> > > > **Thanks for your responses**
> > > >
> > > > I sincerely appreciate the authors' helpful responses, which have effectively addressed most of my concerns. I am leaning towards accepting. However, I prefer to hold my score at this time with increased confidence.

---

> > > > > ### Author Response · Authors · 2023-11-23
> > > > >
> > > > > Dear Reviewer nfgg:
> > > > >
> > > > > Thank you once again for the time and effort you dedicated to reviewing our work. Your support and comments are greatly appreciated.

---

### Official Review · Reviewer_cEiK · 2023-11-04

**Soundness:** 3 good
**Presentation:** 3 good
**Contribution:** 3 good
**Rating:** 6
**Confidence:** 4

**Summary:**

This paper proposes several techniques for improved training of diffusion recovery likelihood models, such as
- learning an initializer model for MCMC,
- noise variance reduction for reducing gradient variance.

This paper shows application of the proposed model to
- conditional generation and classifier-free guidance,
- compositional generation,
- OOD detection.

**Strengths:**

- The paper is clearly written and easy to read.
- CDRL shows clear performance improvements over previous EBM-based generative models.
- CDRL can be applied to tasks such as unconditional generation, conditional synthesis, likelihood estimation, OOD detection, composition, etc. Such tasks were unexplored in the original DRL paper [1].

[1] Learning Energy-Based Models by Diffusion Recovery Likelihood, Gao et al., ICLR, 2021.

**Weaknesses:**

I am inclined to give the score "marginal accept" for the following reasons.
- The proposed method is a straightforward combination of two known techniques, DRL [1] and MCMC amortization [2,3]. While the simplicity of the idea is practically appealing, the idea lacks theoretical novelty. Moreover, the tasks demonstrated in the paper are already well-explored in the diffusion model literature, and the algorithms for the tasks are straightforward extensions of diffusion-based ones, as the score is just the gradient of the energy. For instance, classifier-free guidance is explored in [4], and compositional generation is explored in [5].
- The paper lacks a comparison of inference time for CDRL and the baselines.
- The paper lacks a comparison of training cost (e.g., required VRAM) for CDRL and the baselines. I expect CDRL training is more expensive than diffusion or DRL training, as the former requires two networks while latter only one.
- The paper lacks results on higher resolution (e.g., 256x256) images.

[1] Learning Energy-Based Models by Diffusion Recovery Likelihood, Gao et al., ICLR, 2021.

[2] Approximate Inference with Amortized MCMC, Li et al., 2017.

[3] Learning Energy-Based Prior Model with Diffusion-Amortized MCMC, Yu et al., NeurIPS, 2023.

[4] Classifier-Free Diffusion Guidance, Ho et al., 2022.

[5] Reduce, Reuse, Recycle: Compositional Generation with Energy-Based Diffusion Models and MCMC, Du et al., ICML, 2023.

**Questions:**

- I am curious about the authors' opinion on the practical benefits of CDRL vs. diffusion. Specifically, tasks that can be achieved with CDRL can also be achieved by diffusion, and vice versa, using the relation that the gradient of the energy is the score. I also observe that hyper-parameter choices in this paper are heavily influenced by those of diffusion models. Moreover, while CDRL uses fewer noise levels than diffusion models (e.g., 1000 levels), recent works on fast diffusion sampling have reduced sampling time for diffusion significantly (e.g., 2.87 FID on CIFAR10 with NFE=20 [1]). I think this naturally leads us to wonder what are the strengths of CDRL compared to diffusion.

[1] DPM-Solver: A Fast ODE Solver for Diffusion Probabilistic Model Sampling in Around 10 Steps, Lu et al., NeurIPS, 2022.

---

> ### Author Response · Authors · 2023-11-22
> **Thank you for your comments and review! (1/3)**
>
> Thank you for your comments and review. Here are our responses, hope that can solve your concerns:
>
>
> ### 1. The proposed method is a straightforward combination of two known techniques, DRL and MCMC amortization ....The tasks demonstrated in the paper are already well-explored in the diffusion model literature....
>
> Thank you for your question. The primary aim and contribution of our work is advancing the study of EBM. Although EBMs are gathering increasing attention, scaling up their training remains largely unexplored. The key challenge lies in developing a more efficient sampling strategy. While DRL [1] made strides by using multiple noise levels to ease sampling, it established a strong baseline but still falls short when compared to other generative frameworks. Narrowing this gap has proven challenging, raising questions about the potential of EBMs as generative models. While amortized inference is a known technique in prior studies [2, 3], our work is the first to show that integrating amortized sampling with the DRL framework significantly enhances EBM performance. Concurrent effort [3], explores amortizing MCMC in a EBMs fitted within the latent space of a very simple VAE model. The EBM structure as well as the VAE latent space are not very complex to model. Their focus is different and they require running a 100-step diffusion model alongside 30 or 50 steps of MCMC. In contrast, our model addresses more complex data distributions with just a single-step initialization proposal and 15-step MCMC sampling, achieving much more superior sample quality compared to [3]. This underscores that our scaling-up effort is far from trivial. Regarding tasks like classifier guidance and compositional generation, we do not claim to be the first to explore these areas. Instead, our focus is to demonstrate that our model, while delivering strong performance, is compatible with these techniques, broadening its applicability in diverse applications.
>
> Furthermore, we make many practical design choices related to noise scheduling, network architecture, noise variance reduction. These choices helps the training of EBM and we carefully conducted ablation studies on each component (please our Appendix G). We believe these designs can greatly benefit future works in the study of EBM.
>
> Besides, we show that EBM trained with CDRL can benefit from simple sampling adjustment techniques to greatly reduce its sampling steps. This further improves the sampling efficiency of EBM as a generative model and might inspire future works in similar directions.
>
> Thus, we believe that our work has made great contributions. Our model and algorithm can pave the way for the scaling up of EBM training on complex distributions.
>
> [1] Learning Energy-Based Models by Diffusion Recovery Likelihood, Gao et al., ICLR, 2021.
>
> [2] Approximate Inference with Amortized MCMC, Li et al., 2017.
>
> [3] Learning Energy-Based Prior Model with Diffusion-Amortized MCMC, Yu et al., NeurIPS, 2023.
>
> [4] Classifier-Free Diffusion Guidance, Ho et al., 2022.
>
> [5] Reduce, Reuse, Recycle: Compositional Generation with Energy-Based Diffusion Models and MCMC, Du et al., ICML, 2023.
>
> &nbsp;
> &nbsp;
>
> ### 2. The paper lacks a comparison of inference time for CDRL and the baselines.
> Please refer to the below comparison of our CDRL with other strong EBM baselines. Our CDRL, employing 90 steps (6 x 15), achieves high-quality generation with a relatively short sampling time. Furthermore, as discussed in Section 4.2 of our paper, a simple adjustment in the sampling process can be applied to further reduce the sampling duration without significantly impacting the quality of the generated samples. This comparison is presented in Table 9 of our Appendix.
>
>
> Table: Time for generating 100 Cifar10 samples
>
> | Model Name |  Total MCMC steps | Time (s) | FID |
> |:----------:| ----------------- |------|-----|
> |CoopFlow| 30 | 2.50|15.80|
> |VAEBM| 12 |21.3|12.16|
> |DRL| 6 x 30 = 180 |23.9|9.58|
> |CDRL| 6 x 15 = 90 |12.2|4.31|
> |CDRL (8 steps)| 6 x 8 = 48 |6.48|4.58|
> |CDRL (5 steps)| 6 x 5 = 30 |4.14|5.37|
> |CDRL (3 steps)| 6 x 3 = 18 |2.60|9.67|

---

> ### Author Response · Authors · 2023-11-22
> **Thank you for your comments and review! (2/3)**
>
> ### 3. The paper lacks a comparison of training cost (e.g., required VRAM) for CDRL and the baselines. I expect CDRL training is more expensive than diffusion or DRL training, as the former requires two networks while latter only one.
>
> As detailed in Appendix A2, we utilized 8 A100 GPUs to train our model over 400k iterations, spanning approximately six days. Compared to DRL, although CDRL includes an additional initializer model. As outlined in Appendix A.1., we only employ a small U-Net as the initializer, resulting in the MCMC sampling process predominantly dictating the sampling time. Notably, with CDRL using only half the MCMC sampling steps at each noise level, it requires fewer computational resources than DRL.
>
> More specifically, DRL’s standard configuration, which involves a 6-noise schedule and 30 Langevin steps, demands 240,000 iterations. In contrast, CDRL, utilizing half the Langevin steps and extending over 400,000 iterations, reduces training time to just 80% of that required for DRL. It's also important to note that even with increased computational investment in DRL, such as longer training durations or more MCMC steps, significant performance enhancements remain elusive (as demonstrated in Table 2 of DRL). In comparison, CDRL achieves superior performance with less computational expenditure, demonstrating its efficiency advantage over DRL.
>
> &nbsp;
>
> ### 4. The paper lacks results on higher resolution (e.g., 256x256) images.
> In terms of handling high-resolution images, the recent trend in generative modeling involves either utilizing the latent space of a VAE, as in Latent Diffusion[5], or initially generating a low-resolution image and then scaling up, as demonstrated by techniques like Imagen. This process might reduce the modeled space to dimensions such as 32 × 32 or 64×64, aligning with the resolution on which we conducted our experiments. Following your suggestion, we have carried and extra experiment on CelebA-HQ dataset (256 x 256) in the latent space of latent diffusion.  Our preliminary results have achieved an FID score of 10.74. We have included the new results in our modifed paper (see Figure 9 and Table 7 in the Appendix). In the following Table, we compare our preliminary results with some baselines on this dataset.
>
> Table FID score for CelebA-HQ (256 x 256) dataset
> |Model | FID score|
> |------| ---------|
> |GLOW[3]| 68.93|
> |VAEBM[1] |20.38|
> |ATEBM [2]|17.31 |
> |VQGAN+Transformer[4]| 10.2|
> |LDM[5]|5.11|
> |CDRL(ours)  |10.74|
>
> [1] VAEBM: A Symbiosis between Variational Autoencoders and Energy-based Models
>
> [2] Learning energy-based models with adversarial training
>
> [3] Glow: Generative Flow with Invertible 1x1 Convolutions
>
> [4] Taming Transformers for High-Resolution Image Synthesis
>
> [5] High-Resolution Image Synthesis with Latent Diffusion Models

---

> ### Author Response · Authors · 2023-11-22
> **Thank you for your comments and review! (3/3)**
>
> ### 5. I am curious about the authors' opinion on the practical benefits of CDRL vs. diffusion. Specifically, tasks that can be achieved with CDRL can also be achieved by diffusion, and vice versa, using the relation that the gradient of the energy is the score. I also observe that hyper-parameter choices in this paper are heavily influenced by those of diffusion models. Moreover, while CDRL uses fewer noise levels than diffusion models (e.g., 1000 levels), recent works on fast diffusion sampling have reduced sampling time for diffusion significantly (e.g., 2.87 FID on CIFAR10 with NFE=20 [1]). I think this naturally leads us to wonder what are the strengths of CDRL compared to diffusion.
>
> Thank you for your question. There is a close connection between EBM and diffusion models. Both frameworks offer scores to guide the generation process, whether through Langevin dynamics or SDE/ODE solvers. The key distinction lies in their implementation: EBM models the log-likelihood, whereas diffusion models focus on the gradient of the log-likelihood. We appreciate the significant success achieved by diffusion models. But we believe EBMs also demonstrate substantial applicability, as evidenced by numerous use cases. For example, in composition generation tasks, one study [3] highlights the benefits of using an energy function with advanced samplers like HMC to achieve improved results. In Simulation-Based Inference, another research [4] employs a fitted function for neural likelihood estimation, and continuous learning applications [5] leverage the energy function for model fusion. Furthermore, EBMs are used to model set functions [6], showcasing their versatility across various domains.
>
> We believe in the untapped potential of EBMs, either as robust generative models or in applications requiring complex distribution likelihood estimation. However, modeling intricate distributions like images remains a challenge for EBMs, limiting their performance in generation and likelihood estimation. This limitation largely stems from the absence of efficient sampling strategies for stable training. Our research advances EBM studies by devising a simple yet effective training methodology. The inherent link between EBM and diffusion models enables us to adopt advantageous techniques from diffusion models, like noise level design, to improve EBM training. Nonetheless, specific aspects of EBM also require careful consideration, including Langevin sampling parameters, the EBM network architecture, effective initializers, and noise variance reduction techniques. We have implemented several practical approaches to enhance EBM training.
>
> We appreciate the successes of rapid ODE solvers, such as [1]. However, as detailed in Section 4.2 of our paper, EBM sampling also shows potential for acceleration through sampling adjustment techniques. Our preliminary results, achieved with a simple strategy that further reduces MCMC sampling steps to 20 or 30, are promising. We believe that continued research in this area could lead to the development of a more sampling-efficient model.
>
>
> [1] DPM-Solver: A Fast ODE Solver for Diffusion Probabilistic Model Sampling in Around 10 Steps, Lu et al., NeurIPS, 2022.
>
> [2] Learning energy based models by diffusion recovery likelihood.
>
> [3] Reduce, Reuse, Recycle: Compositional Generation with Energy -Based Diffusion Models and MCMC.
>
> [4] Maximum Likelihood Learning of Energy-Based Models for Simulation-Based Inference.
>
> [5] Beef: Bi-compatible class-incremental learning via energy-based expansion and fusion.
>
> [6] Learning Neural Set Functions Under the Optimal Subset Oracle.

---

> > ### Comment · Reviewer_cEiK · 2023-11-22
> >
> > Thank you for the detailed feedback! I now see that EBMs may be more practical than diffusion models in some tasks, as written in the authors' answer to my question. However, I feel that the paper could have been better if the authors had compared CDRL with diffusion in the mentioned tasks. Also, I don't think results on CelebA-HQ $256 \times 256$ in the **latent space of LDM** demonstrates the scalability of CDRL. The scalability in LDM+CDRL arises mostly from the autoencoder in the LDM, rather than CDRL itself, since CDRL only sees $32 \times 32$ or $64 \times 64$ resolution images in this setting. Weighing the pros and cons of CDRL, I will keep my current score, which is already positive.

---

> ### Author Response · Authors · 2023-11-23
> **Thank your for your comments！**
>
> Dear Reviewer cEiK:
>
> Thank you for your comments. We truly appreciate your valuable feedback and the overall positive attitude you have taken towards our paper. In terms of generating high-resolution images, it would indeed be interesting to further advance CDRL for high-resolution generation in the pixel space. We choose to start from training CDRL in the latent space of LDM because from our observation, the success of recent diffusion-based generative models [1, 2] suggests that compared to having a single model handling high resolution generation directly, it is more parameter-efficient and training-efficient to either build a cascaded pipeline in the pixel space or build the model in the latent space of a VAE. For both cases, the base model is still operated on low-resolution images or latents. For the cascaded pipeline, the super-resolution models are easier to learn, and usually require fewer number of parameters than the base model. We are keen to further expolring this generation pipeline with CDRL and applying CDRL framework to real world applications and science discoveries in future work.
>
> [1] High-Resolution Image Synthesis with Latent Diffusion Models
> [2] Photorealistic Text-to-Image Diffusion Models with Deep Language Understanding

---

> ### Author Response · Authors · 2023-11-23
> **Comparison between EBM and Diffusion in OOD detection**
>
> While undertaking a completely new experiment during the discussion period might be challenging, we might delv deeper into the task of Out-of-Distribution (OOD) detection. We searched for diffusion based models in this task and currently we found the following 2 related works[1, 2]. These studies indicate that diffusion can perform reconstruction-based OOD detection. The process starts by degrading the input image, either by injecting Gaussian noise to the whole image, as done in [1], or by masking a part of the image, as in [2]. The degraded image is then reconstructed using diffusion process. The reconstruction similarity score (such as LPIPS), calculated between the input and reconstructed images, serves as the metric for OOD detection. On the other hand, as previously discussed, EBM estimate the log-likelihood directly as the energy function, allowing them to utilize this energy for OOD detection, along with the reconstruction method, thus offering more possibilities for this task.
>
> In reviewing the results from [1, 2], we noticed an interesting observation. The reconstruction method seems more effective at detecting OOD samples that are significantly different from the training samples. For instance, it's more successful when using CIFAR-10 as in-distribution samples and SVHN as OOD samples (with AUROC scores of 0.979 in [1] and 0.992 in [2]). However, it might struggle with OOD samples that are similar to the training distribution, as evidenced by the lower AUROC scores for CelebA OOD samples in [1] (score 0.685) and CIFAR-100 in [2] (score 0.607).  As a comparison, using the estimated likelihood as detection score, CDRL achieves 0.78 on Cifar100, 0.84 for CelebA and 0.82 for SVHN. We also make a preliminary test for CDRL on SVHN as OOD dataset using the reconstrution based method simialr to [1]. Without too much tuning for the hyper-parameters, CDRL achieve a score of 0.962, which is comparible to [1]. This means CDRL can also do reconstruction based OOD detection.
>
> The results imply that both reconstruction-based and likelihood-based methods might have their distinct advantages and limitations depending on the dataset. Compared to diffusion-based models, EBMs offer more possibilities as they are capable of using both methods. Furthermore, with certain designs of the energy function, using likelihood-based method, [3] seems to perform well on multiple datasets .
>
>
> [1] Denoising diffusion models for out-of-distribution detection
>
> [2] Unsupervised Out-of-Distribution Detection with Diffusion Inpainting
>
> [3] Guiding Energy-based Models via Contrastive Latent Variables.

---

### Official Review · Reviewer_9dWK · 2023-11-06

**Soundness:** 3 good
**Presentation:** 3 good
**Contribution:** 2 fair
**Rating:** 8
**Confidence:** 5

**Summary:**

This paper proposes a novel training algorithm for jointly training an energy-based model (EBM) and an initializer model. The initializer has a very similar form to DDPM, so the proposed method can be viewed as a cooperation of EBM and DDPM.

**Strengths:**

* The overall exposition is clear to follow.
* The proposed algorithm, CDRL, shows a clear improvement over Diffusion Recovery Likelihood (DRL), an algorithm that CDRL is based on. The sample quality (in FID) is improved while the required computation (in MCMC steps) is reduced.
* CDRL demonstrates broad applicability over multiple tasks outside image generation, such as compositional generation and out-of-distribution detection.

**Weaknesses:**

* The empirical performance of CDRL is good but not very strong. It is clear that CDRL is an improvement over DRL, but it still falls behind other models in multiple tasks.
    * The initializer is very similar to DDPM, but CDRL's unconditional CIFAR-10 FID is worse than DDPM (Table 1).
    * Also in Table 3, CDRL is outperformed by DDPM++.
* In the out-of-detection distribution (OOD) experiment (Table 4), OOD detection capability is evaluated on only three test datasets, and the number of test OOD datasets is too small. It is important to use diverse test OOD datasets in evaluation because an OOD detector needs to detect any possible outliers. The current practice typically uses at least five OOD datasets that have different visual characteristics. Also, SVHN is known to be a highly challenging OOD dataset, particularly for generative models [1], but not included in Table 4. It would be great to see CDRL is able to detect SVHN as OOD.
* It would be nice if the authors could comment on the consistency of the model after adding the initializer model. Would it alter the consistency?

[1] Nalisnick et al., Do Deep Generative Models Know What They Don't Know?, https://arxiv.org/abs/1810.09136

**Questions:**

See weaknesses.

===
The authors have addressed my questions in detail.

---

> ### Author Response · Authors · 2023-11-22
> **Thank your for you constructive comments and review! (1/2)**
>
> Thank you for your reviews and comments. We find they are constructive to our paper. Please see our responses below. Hope they can solve your concerns.
>
> ### 1. The empirical performance of CDRL is good but not very strong. It is clear that CDRL is an improvement over DRL, but it still falls behind other models in multiple tasks.
>
> Thank you for your question. The primary objective of our research is to push forward the study of EBM. As we discuss in our paper, EBMs have gathered increasing interest in recent research. However, unlike the widely-studied diffusion models, scaling up EBMs remains an area less explored. Prior studies like DRL have adopted noise level concepts from diffusion models into EBM, setting a strong baseline. Despite this, a significant performance gap remains when compared to diffusion models. Our work builds upon DRL, aiming to narrow this gap.
>
> Our initializer, while drawing inspiration from DDPM's design, serves a distinct purpose. We employ the initializer only for providing an initial rough estimate at each step, with the subsequent generation process guided by MCMC under the EBM framework. This approach is illustrated in (current) Figure 8 of our Appendix. As a result, we utilize a smaller initializer, which is also cooperatively learned from the revised samples of EBM. The difference between learning directly from data and this cooperative learning approach is detailed in Appendix G.2.
>
> Furthermore, we have implemented other practical strategies for EBM training. Our findings show that the sampling efficiency of CDRL can be greatly improved using simple sampling techniques. While we acknowledge the successes of diffusion models and recognize the current performance disparity with energy-based models, we still believe in the potential of EBMs as powerful generative models and in estimating the likelihood of complex distributions. Our work significantly advances the generative capabilities of EBM.
>
> &nbsp;
>
> ### 2. It would be nice if the authors could comment on the consistency of the model after adding the initializer model. Would it alter the consistency?
>
> In our framework, the EBM is estimated by maximimizing recovery likelihood, where the samples are generated by starting a few MCMC sampling steps from samples given by the initializer model. It remains an unbiased estimator of the true model parameters, which means given enough data, a rich enough EBM and exact synthesis, it learns $p_\theta = p_{\rm data}$. On the other hand, the initializer model is learned by maximizing log-likelihood, treating the samples from the EBM as the training data. Therefore, it is an unbiased estimator of the EBM model distribution $p_\theta$.

---

> ### Author Response · Authors · 2023-11-22
> **Thank your for you constructive comments and review! (2/2)**
>
> ### 3. In the out-of-detection distribution (OOD) experiment (Table 4), OOD detection capability is evaluated on only three test datasets, and the number of test OOD datasets is too small. It is important to use diverse test OOD datasets in evaluation because an OOD detector needs to detect any possible outliers. The current practice typically uses at least five OOD datasets that have different visual characteristics. Also, SVHN is known to be a highly challenging OOD dataset, particularly for generative models [1], but not included in Table 4. It would be great to see CDRL is able to detect SVHN as OOD.
>
> Thank you for raising this question. Your insights have motivated us to conduct a more thorough investigation during the rebuttal phase. In response to your suggestions, we have conducted experiments on five different datasets and incorporated more recent EBM baselines. Our experimental findings are detailed in the table below. (This table is included as Table 8 in the current modified version of our paper). Key observations include:
>
> 1. Comparing with other 3 datasets, SVHN and Texture are indeed more challenging. Our CDRL makes valid prediction on these datasets (0.82 on SVHN 0.65 on Texture).
> 2. CDRL consistently outperforms DRL in most datasets and matches its performance on SVHN. This demonstrates CDRL's enhancement of DRL in energy estimation.
> 3. While CDRL may not achieve state-of-the-art results on all datasets comparing to baselines like [2], as discussed in [2], the addition of a contrastive term to the energy function, significantly benefits OOD detection. Considering that our proposed CDRL is a framework designed to stabilize EBM training and is compatible with any energy function design (we only need to include multiple noise levels and an initializer at each level), we anticipate improved performance with an energy design incorporating a contrastive loss similar to [2].
>
> We sincerely appreciate your query as it has spurred us to deepen our study of the OOD task.
>
> Table 1: OOD results using Cifar 10 as in distribution sample
>
> |                | Cifar-10 interpolation | Cifar-100 | CelebA | SVHN | Texture |
> |----------------|------------------------|-----------|--------|------|---------|
> | PixelCNN       | 0.71                   | 0.63      | -      | 0.32 | 0.33    |
> | GLOW           | 0.51                   | 0.55      | 0.57   | 0.24 | 0.27    |
> | NVAE           | 0.64                   | 0.56      | 0.68   | 0.42 | -       |
> | IGEBM          | 0.70                   | 0.50      | 0.70   | 0.63 | 0.48    |
> | VAEBM          | 0.70                   | 0.62      | 0.77   | 0.83 | -       |
> | EBM CD         | 0.65 (-)               | 0.83 (0.53)| - (0.54)| 0.91 (0.78)| 0.88 (0.73) |
> | CLEL           | 0.72                   | 0.72      | 0.77   | 0.98 | 0.94    |
> | DRL \*         | -                      | 0.44      | 0.64   | 0.88 | 0.45    |
> | MPDR-S         | -                      | 0.56      | 0.73   | 0.99 | 0.66    |
> | MPDR_R         | -                      | 0.64      | 0.83   | 0.98 | 0.80    |
> | CDRL           | 0.75                   | 0.78      | 0.84   | 0.82 | 0.65    |
>
> &nbsp;
>
> Note: 1. \* the score for DRL is reported by [1]; 2.For EBM CD, we find different numbers from different sources, one from the orginal paper[3] and the other from a recent work[1], we include both scores here and put the scores from [1] into brackets.
>
> &nbsp;
>
> [1] Energy-Based Models for Anomaly Detection: A Manifold Diffusion Recovery Approach
>
> [2] Guiding Energy-based Models via Contrastive Latent Variables.
>
> [3] Improved contrastive divergence training of energy-based models.

---

> ### Comment · Reviewer_9dWK · 2023-11-22
> **Thank you for your response**
>
> I have read the author's response and sincerely appreciate their thorough answer.
>
> Most of my concerns are addressed, and therefore I would love to increase my score.
>
> Even though the proposed method does not show state-of-the-art generation and out-of-distribution detection performance, I believe this paper is indeed a meaningful contribution to energy-based model research.

---

> ### Author Response · Authors · 2023-11-23
> **Thank you for your feedback!**
>
> Dear Reviewer 9dWK:
>
> Thank you for your insightful review and query. Your constructive feedback offers valuable insights that helps us make a better paper.  We are sincerely grateful for your recognition of our contribution to EBM research. Additionally, we really appreciate your decision to raise your score. We will remain committed to advancing the performance of EBM in both generation and OOD tasks.

---

### Meta-Review · Area_Chair_u3Qu · 2023-12-05

**Metareview:**

The paper proposes a technical improvement to energy-based generative model (EBM) based on the diffusion recovery likelihood (DRL) formulation: a sample initializer is introduced to accelerate convergence of energy-model-driven MCMC in each denoising step, which is learned along with the training of the energy models. The presentation is neat, and the method seems solid. The method achieves seemingly state-of-the-art image generation performance among EBMs, and the benefits/utilities of classifier-free guidance, compositionality and OOD detection are demonstrated. Nevertheless, the proposed method is somehow specific to the DRL-based EBMs, and a few reviewers mentioned insufficient ideological innovation. The relative improvement over existing EBMs comes with the requirement of training an additional model. Given the resemblance to diffusion models, it is noted that the absolute performance does not surpass state-of-the-art diffusion models, and it is expected to discuss more about the relation with diffusion models (especially the resemblance to the predictor-corrector paradigm in diffusion models).

**Justification For Why Not Higher Score:**

The mentioned weaknesses may limit the interest and impact to a broader level.

**Justification For Why Not Lower Score:**

The work exploited the potential of EBMs for image generation, and showed encouraging results that reach the level of quality of diffusion models. All reviewers acknowledged this contribution.

---

### Decision · Program_Chairs · 2024-01-16

Accept (spotlight)